# 3D reconstruction from cryo-EM projection images using two spherical embeddings

Yonggang Lu [1✉], Jiaxuan Liu [1], Li Zhu [2,3✉], Bianlan Zhang[1] & Jing He[4]

Single-particle analysis (SPA) in cryo-electron microscopy has become a powerful tool for determining and studying the macromolecular structure at an atomic level. However, since the SPA problem is a non-convex optimization problem with enormous search space and there is high level of noise in the input images, the existing methods may produce biased or even wrong final models. In this work, to deal with the problem, consistent constraints from the input data are explored in an embedding space, a 3D spherical surface. More specifically, the orientation of a projection image is represented by two intersection points of the normal vector and the local X-axis vector of the projection image on the unit spherical surface. To determine the orientations of the projection images, the global consistency constraints of the relative orientations of all the projection images are satisfied by two spherical embeddings which estimate the normal vectors and the local X-axis vectors of the projection images respectively. Compared to the traditional methods, the proposed method is shown to be able to rectify the initial computation errors and produce a more accurate estimation of the projection angles, which results in a better final model reconstruction from the noisy image data.

[1] School of Information Science and Engineering, Lanzhou University, Lanzhou, Gansu 730000, China. [2] School of Life Sciences, Lanzhou University, Lanzhou, Gansu 730000, China. [3] Electron Microscopy Centre of Lanzhou University, Lanzhou, Gansu 730000, China. [4] Department of Computer Science, Old Dominion University, Norfolk, VA 23529-0162, USA. ✉email: ylu@lzu.edu.cn; zhuli@lzu.edu.cn

Cryo-electron microscopy (Cryo-EM) has become an increasingly powerful and popular technique for three-dimensional (3D) structure determination of biological macromolecules[1,2]. With recent revolutionary technological advancements, single-particle analysis (SPA) in cryo-EM has been used to successfully determine the structure of biological macromolecules to near atomic resolution[3–6]. Cryo-EM was selected by the journal *Nature Methods* as the Method of the Year 2015[2]. The advances in the cryo-EM technique were also recognized by the 2017 Nobel Prize in Chemistry (https://www.nobelprize.org/nobel_prizes/chemistry/laureates/2017).

In SPA, the 3D structure of a molecule is reconstructed from a large number of its 2D projection images taken by a transmission electron microscope (TEM). The most important and challenging task in the 3D structure reconstruction is to determine the orientation of individual 2D experimental projection images, which are usually represented by projection angles[7,8]. Nevertheless, due to the low signal-to-noise ratio (SNR) in cryo-EM, contrast transfer function (CTF) of the microscope, and errors introduced during particle picking, it is usually difficult to determine the projection angles directly from the 2D projection images[9]. Thus, most of the existing 3D reconstruction methods use expectation maximization to iteratively refine the projection angle estimation and the 3D model starting from an initial condition, such as in EMAN[10] and RELION[11]. Improvements of the methods include using stochastic hill climbing[12,13], stochastic gradient descent[14], and particle filtering with a posterior probability density estimation[15]. Although in some cases these methods can converge successfully from a random initialization, they may still converge to local optimal solutions[16,17].

The 3D reconstruction problem can be viewed as an optimization problem in which a 3D model is searched to agree with 2D projection images overall. The major difficulty is that it is a non-convex optimization problem with enormous search space, which implies that the initialization may affect the optimization greatly[16]. The impacts of the initialization on the 3D reconstruction have been observed in many experiments. With low-quality initial models, biased final models[13,18] or even completely wrong reconstructions, such as the Einstein from noise pitfall[19] may be produced. The situation becomes worse when the 3D model contains no symmetry. Consequently, there is still great need for developing robust 3D reconstruction methods.

A plausible way towards this goal is to use priors as little as possible while exploring more constraints from the input data. One traditional solution is to estimate the orientation of the projection image using angular reconstitution[20], based on the estimated common lines among the projection images according to the central section theorem[21]. The methods based on angular reconstitution can be used to estimate the projection angles without using any prior information. However, these traditional methods rely heavily on the common line detection between projection images, while low-quality common lines are usually estimated from the noisy input images, which can affect the reconstruction result greatly[13,19]. Moreover, the estimated common lines are usually inconsistent among different image pairs, which cannot be satisfied together during the 3D reconstruction.

In order to build more robust common line-based 3D reconstruction methods, consistent information needs to be extracted from the input of inconsistent common lines. Singer et al. have proposed methods to enhance the accuracy of orientation determination by using consistency constraints of all the common lines[22–24]. In 2010, they have proposed a voting algorithm[22] which uses an improved Bayesian approach based on voting to identify consistent common line information and produces a voting score which represents the reliability of each common line. The voting method can be used to estimate the common lines of the projection images, but cannot be used to directly estimate the projection angles.

Later, they have proposed two methods for simultaneous projection angle estimation considering all the common lines at once. One is called the least unsquared deviations (LUD) method[23], which first defines a global self-consistency error of all common lines and then a global orientation assignment is produced by minimizing a global self-consistency error via semi-definite relaxation. The other is called the synchronization method[24] which uses common lines between all the triplets of projection images to define spatial constraints. The global orientation assignment is produced by maximizing the number of the satisfied spatial constraints, which is solved by transferring it into an eigenvalue problem of an appropriately constructed matrix. Later, the second method is improved by first partitioning the images into two consistent handedness groups followed by handedness synchronization[25]. Both the LUD method and the synchronization method estimate all the projection angles simultaneously by maximally satisfying the constraints extracted from all the common lines together, thus it can work with falsely detected common lines under low SNR.

In this work, instead of using the consistency constraints of all the common lines, the consistency constraints of the relative orientations (relative projection angles) of all the projection images are exploited. In our method, the orientation of a projection image is represented by its normal vector and its local X-axis vector which are perpendicular to each other in the 3D space as shown in Fig. 1. It can be seen from Fig. 1b that the normal vector of a projection image can be determined by the Euler angles $\alpha$ and $\beta$, and vice versa. So, the normal vector is equivalent to the Euler angles $\alpha$ and $\beta$. When the normal vector is given, the local X-axis vector can be used to determine the in-plane rotation, which gives the Euler angle $\gamma$. As shown in Fig. 1b, the normal vector and the local X-axis vector can be represented by their intersection points on a 3D unit spherical surface, respectively. This way, the orientation of a projection image can be represented by two intersection points on the unit 3D spherical surface. It can be seen that the 3D spherical surface is a natural space for representing the orientation of the projection images.

To represent the relative orientation between two projection images, the dihedral angle and the angle between the local X-axis vectors of the two images can be used. The dihedral angle is the angle between the normal vectors of the two images, which is also equivalent to the distance between the two intersection points of the two normal vectors on the unit spherical surface.

In the proposed method, the dihedral angles and the angles between X-axis vectors are estimated using two spherical embeddings separately. First, the dihedral angles between all the pairs of projection images are estimated from the common lines using the voting method[22] introduced by Singer et al. in 2010. The estimated dihedral angle between a pair of images is produced by considering not only the common line of the two images, but also the common lines between an image in the pair and all the other images, so it is more reliable than the common line information. However, the estimated dihedral angles are still not consistent between all the image pairs in the 3D space, because they are estimated independently. Theoretically, in order to satisfy the independently estimated dihedral angles simultaneously, the degree of freedom has to be conserved. If there are $N$ (>3) projection images, the independently estimated dihedral angles (angle between normal vectors) should have the same degree of freedom as the $N$ normal vectors of the $N$ images. Since the degree of freedom of the $N$ normal vectors equals $N$, the independently estimated dihedral angles can be satisfied simultaneously in an $N$-dimensional space, but not in a 3D space. So, by reducing the dimensionality from $N$ to 3 using a

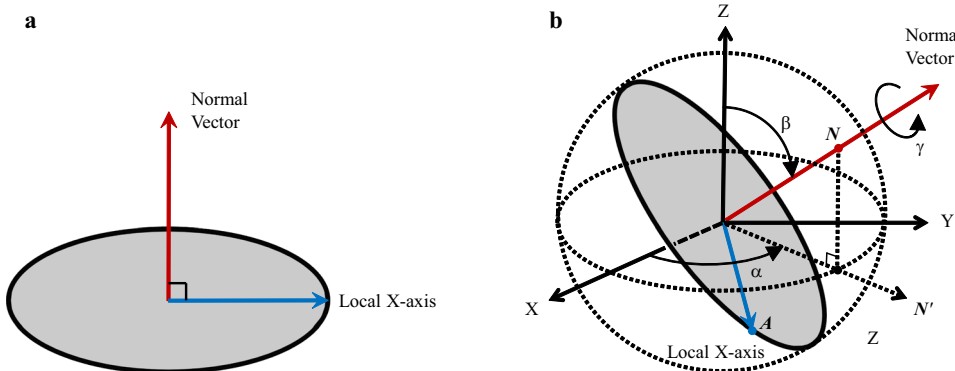

**Fig. 1 The orientation of a projection image can be represented by its normal vector and its local X-axis vector which are perpendicular to each other.**
**a** The projection image in its local coordinate system. **b** The projection image in the 3D reconstruction space, where the normal vector can be represented by its intersection point $N$ on the unit spherical surface, and the local X-axis vector can be represented by its intersection point $A$ on the unit spherical surface. The Euler angle $\beta$ is the angle between the Z-axis and the normal vector, the Euler angle $\alpha$ is the angle between the X-axis and $N'$ vector which is the projection of the normal vector on the X–Y plane, and the Euler angle $\gamma$ is determined by the in-plane rotation of the local X-axis vector along the normal vector.

dimensionality reduction method, the consistent dihedral angles in the 3D space can be obtained. The dimensionality reduction used in our method is spherical embedding[26] which is an unsupervised machine learning method whose aim is to find a representation of the data on a certain spherical surface such that the distances (or dissimilarities) between the data points are preserved as better as possible. Using the spherical embedding, the intersection points of the normal vectors of the projection images on the 3D unit spherical surface can be determined, which gives the consistent dihedral angles in the 3D space. The consistent angles between X-axis vectors of the projection images in the 3D space can be produced similarly using another spherical embedding. Finally, the results of the two spherical embeddings are aligned together using an orthogonal constraint to produce the final estimation of the orientations for the projection images.

The benefit of using the spherical embedding compared with other methods for dealing with the inconsistency is that the spherical embedding is a dimension reduction method whose results will be determined mainly by the consistent information on the 3D spherical surface which is the natural space for representing the relative orientations, leading to a more accurate and robust projection angle estimation.

## Results

The proposed method (Spherical Embedding) is implemented in Matlab and is compared with the LUD method[23], the Synchronization method with partitioning of handedness groups[25], EMAN 2.1[10], and RELION-2[11] on both simulated projection images and two real projection images. These projection images belong to 3D models with no symmetry.

**Preparation of the simulated projection images**. The simulated projection images are generated from the EM density map of Escherichia coli 70S ribosome-ArfA-RF2 complex (EMD-3508, ref. [27]) with a resolution of 3.1 Å (0.143 FSC). The real map is rotated at random angles from uniform distribution on SO(3) to produce $n$ simulated projection images. Four sets of the clean projection images are produced, setting with $n = 100$, 500, 1000, and 2000, respectively. The box size of each image is $260 \times 260$ pixels, with a pixel size of 1.084 Å. To evaluate the performance of the methods under different noise levels, Gaussian white noise, setting with SNR = 0.1, 0.2, 0.3, 0.4, 0.5, and 1.0, respectively, is applied to the $n = 1000$ image set to produce 6 sets of noisy projection images, which is called ImageSet_A. To evaluate the

performance of the methods with different number of input images under a certain noise level, Gaussian white noise with SNR = 0.2 is added to the four sets of the clean projection images (with $n = 100$, 500, 1000, and 2000) to produce 4 sets of noisy projection images, which is called ImageSet_B. For the reconstructions from the simulated projection images, ImageSet_A and ImageSet_B, the real map EMD-3508 serves as the reference map.

**Evaluation of the estimated projection angles for the simulated data**. The proposed method (Spherical Embedding) is compared with the methods of LUD and Synchronization for the projection angle estimation. The ImageSet_A dataset is used to evaluate the estimated projection angles under different noise levels. For the simulated projections, their actual projection angles are known, which are called the reference angles.

First, the estimated dihedral angles between each pair of the projection images are computed from the estimated projection angles, while the actual dihedral angles are computed from the reference angles. The estimated dihedral angles under different noise levels are plotted versus the actual dihedral angles in Fig. 2. The initial dihedral angles estimated by the voting method[18] are used as the input for the proposed method, and the results of the voting method are also shown in the first column of Fig. 2 for comparison. It demonstrates that with the SNR increasing, all the estimations become more accurate. The proposed method produces the most accurate estimation under all 6 different SNR levels, rectifying the input dihedral angles obtained by the voting method (which produces lots of large errors in the estimation except for the case of SNR = 1.0). In addition, the methods of LUD and synchronization generate more accurate dihedral angle estimation than the voting method, but less accurate estimation than the proposed method. When SNR = 0.1, the proposed method produces dihedral angle estimation with the maximum error = 24.5°, while the other methods all produce much larger estimation errors (the maximum errors for the voting method, the LUD method, and the synchronization method are 162.9°, 71.5°, and 57.4°, respectively). It is also noted that, when SNR ≥ 0.2, the proposed method can produce very accurate results with the maximum estimation error as low as 2.6°. It can be seen that the spherical embedding in the proposed method is very effective for producing accurate estimations of the dihedral angles, which correspond to the Euler angles $\alpha$ and $\beta$.

To evaluate the estimated projection angles corresponding to the Euler angles $\alpha$, $\beta$, and $\gamma$, the estimated projection angles of all

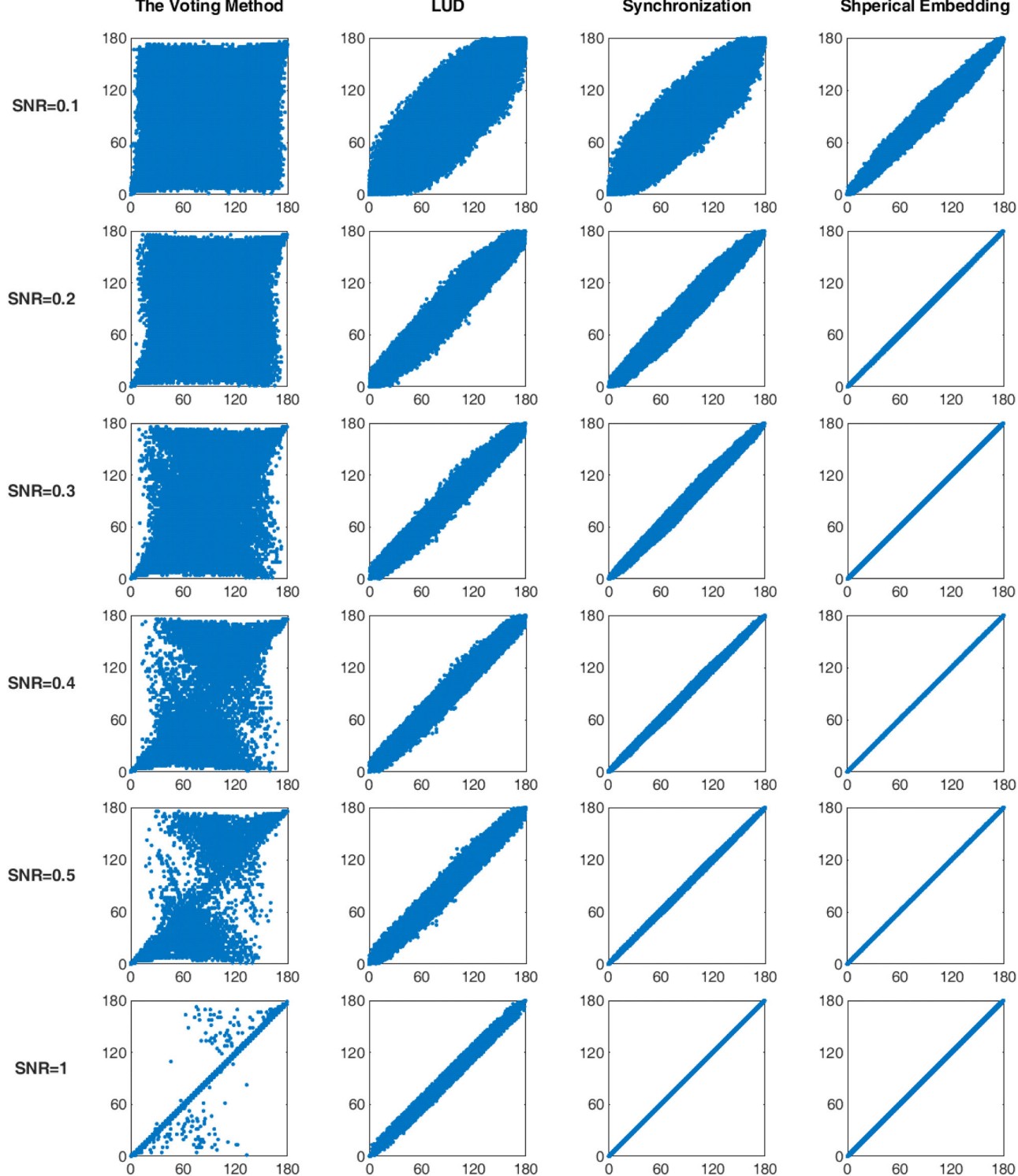

**Fig. 2 Comparison of the estimated dihedral angles (vertical axis) and the actual dihedral angles (horizontal axis) for simulated images produced at different SNR levels (0.1, 0.2, 0.3, 0.4, 0.5, 1.0), using the Voting method, LUD, Synchronization, and the proposed method (Spherical Embedding).**

the images are aligned to the reference angles by a rotation transformation around the center and an optional mirror transformation, to minimize the sum of the differences between the two sets of the projection angles. The root mean square error (RMSE) of the Euler angles $\alpha$, $\beta$, and $\gamma$ between the aligned estimated angles and the reference angles are listed in Table 1. Since the dihedral angles produced by the voting method are not consistent for all the images, they cannot be used to estimate the

projection angles directly. So only the results of the proposed method, the LUD method and the synchronization method are listed in Table 1. It can be seen that except SNR = 1.0, the proposed method produces the lowest RMSEs in all three Euler angle estimations. When SNR = 1.0, all the three methods can produce very good estimation, while the synchronization method produces the best one. When SNR is low, the performance gain of using the proposed method becomes prominent. For example,

**Table 1 Comparison of the RMSEs (in degrees) of the estimated Euler angles produced at different SNR levels by LUD, Synchronization, and the proposed method (Spherical Embedding).**

| SNR | LUD | | | Synchronization | | | Spherical Embedding | | |
|---|---|---|---|---|---|---|---|---|---|
| | $\alpha$ | $\beta$ | $\gamma$ | $\alpha$ | $\beta$ | $\gamma$ | $\alpha$ | $\beta$ | $\gamma$ |
| 0.1 | 10.10 | 23.48 | 21.59 | 10.56 | 13.90 | 15.29 | **2.60** | **4.81** | **6.02** |
| 0.2 | 3.27 | 8.34 | 7.64 | 4.30 | 4.72 | 5.32 | **0.41** | **0.68** | **0.70** |
| 0.3 | 2.14 | 3.51 | 2.92 | 2.05 | 2.05 | 2.25 | **0.26** | **0.53** | **0.50** |
| 0.4 | 1.75 | 2.54 | 2.04 | 1.00 | 1.02 | 1.09 | **0.23** | **0.54** | **0.51** |
| 0.5 | 1.60 | 2.07 | 1.60 | 0.54 | 0.64 | 0.67 | **0.22** | **0.52** | **0.49** |
| 1.0 | 0.47 | 0.77 | 0.65 | **0.20** | **0.48** | **0.47** | 0.22 | 0.52 | 0.50 |

The bold value represent the best result for each Euler angle estimation at a specific SNR level.

when SNR is 0.2 or 0.1, the RMSE produced by the proposed method is much lower than that produced by either LUD or synchronization.

**Evaluation of the 3D reconstruction under different noise levels**. In order to evaluate the performance of the initial model reconstruction from images at different noise levels, 3D reconstruction is produced from the images by different methods without using 2D classification. After the projection angles are estimated using the proposed method (Spherical Embedding), LUD and Synchronization for ImageSet_A, the 3D reconstruction results are produced using the FIRM method in the ASPIRE 0.14 (http://spr.math.princeton.edu) with the maximum iteration number setting to 30. In addition, EMAN 2.1 and RELION-2 are also used to reconstruct the initial models from the images in ImageSet_A. Resolution is estimated by calculating the FSC curve[28] between the reconstructed 3D model and the original EMD-3508 density map. Results from different methods are shown in Fig. 3.

It can be seen from Fig. 3 that the FSC curves produced by the proposed method (Spherical Embedding) are above the curves produced by all the other methods when SNR is set to 0.1, 0.2, 0.3, 0.4, and 0.5. When SNR = 1, the FSC curves produced by the proposed method, the LUD method and the Synchronization method overlaps, while they are all above the curves produced using EMAN 2.1 and RELION-2. The results demonstrate that the proposed method generates the best reconstruction for the 1000 projection images at different SNR levels. Furthermore, when the SNR level is as low as 0.2 or 0.1, the proposed method can still produce obviously higher resolution models compared to the other methods.

**Evaluation of the 3D reconstruction using different number of projection images**. To evaluate the performance of the initial model reconstruction from different number of images, similar experiments are performed as in the above evaluation, except that ImageSet_B is used instead of ImageSet_A. The SNR levels are all 0.2 for the images in ImageSet_B. The FSC curves between the reconstructed 3D model and the original density map of different methods are shown in Fig. 4a. It is noticed that better models can usually be reconstructed from projection images with a larger dataset, while the proposed method (Spherical Embedding) can produce the best model within all the five methods as indicated by the FSC curves. When the number of images is larger than 500, the benefit of using the proposed method becomes prominent.

As a visual example for the reconstructions from the simulated projection images, Fig. 4b through Fig. 4g show the reference map (EMD-3508) and the reconstructions from the simulated data ($n = 1000$, SNR = 0.2) which is a subset of the ImageSet_B dataset. Spherical Embedding, Synchronization, and LUD produce

similar maps with the reference map. Among the three results, the reconstruction by Spherical Embedding is more similar to the reference map in details. Poor results are produced by both EMAN 2.1 and RELION-2. The poor result of EMAN 2.1 may be caused by that the filtering algorithm in EMAN 2.1 is not suitable for the simulated noise. And for RELION-2, the low resolution is related to its initial model reconstruction process.

**Evaluation of the 3D reconstructions for the EMPIAR-10028 dataset**. The first real projection data, micrographs and particle coordinates of the Plasmodium falciparum 80S ribosome dataset, are downloaded from the EMPIAR database (EMPIAR-10028, ref. [29]). The corresponding EM density map EMD-2660 is also downloaded, which is used as the reference map for resolution estimation. The original sampling rate for the deposited dataset of EMPIAR-10028 and EMD-2660 is 1.34 Å/pixel, which is binned by 2 in the down-sampling process for our subsequent procedure, resulting in a pixel size of 2.68 Å. CTFFIND4[30] is performed to determine the CTF parameters for the entire dataset. Totally 136 out of 499 micrographs with defocus values lower than $-2.2\,\mu m$ are selected for particle extraction, yielding a 11,983 particle set, which is subjected to 2D classification by RELION. To obtain enough 2D class averages as input projection images for calculating the initial model by different methods, class averaging process in RELION is run 50 times, and class averages with good quality and low similarity are manually selected, yielding 531 class averages, named ImageSet_C. The ImageSet_C dataset is used as input to generate 3D initial models by Spherical Embedding, Synchronization, LUD, EMAN 2.1, and RELION-2, instead of using the original 11,983 particle images. For Spherical Embedding, Synchronization and LUD, the class averages are aligned by the center alignment algorithm from ASPIRE 0.14 toolkit. While for EMAN 2.1 and RELION-2, they have their own center alignment algorithm in the initial model reconstruction process.

Resolutions are estimated by calculating FSC curves between the reconstructed density maps and the reference map of EMD-2660 binned to 2.68 Å/pixel. The FSC curves are shown in Fig. 5a and Fig. 5b. It can be seen from Fig. 5a that the FSC curves produced by Spherical Embedding, Synchronization, and LUD are very similar to each other, while they are all much better than the curves produced by EMAN 2.1 and RELION-2. From Fig. 5b which shows the details within the dashed red rectangle in Fig. 5a, it can be seen that the resolution of the map produced by the Spherical Embedding is slightly better than both Synchronization and LUD. The performance gain of the proposed method on this dataset is not as remarkable as on ImageSet_A and ImageSet_B, which may be caused by the high SNR and the small number of the class averages in ImageSet_C.

Figure 5c through Fig. 5h show the reference map (EMD-2660) and the reconstructions by the five methods. Spherical Embedding,

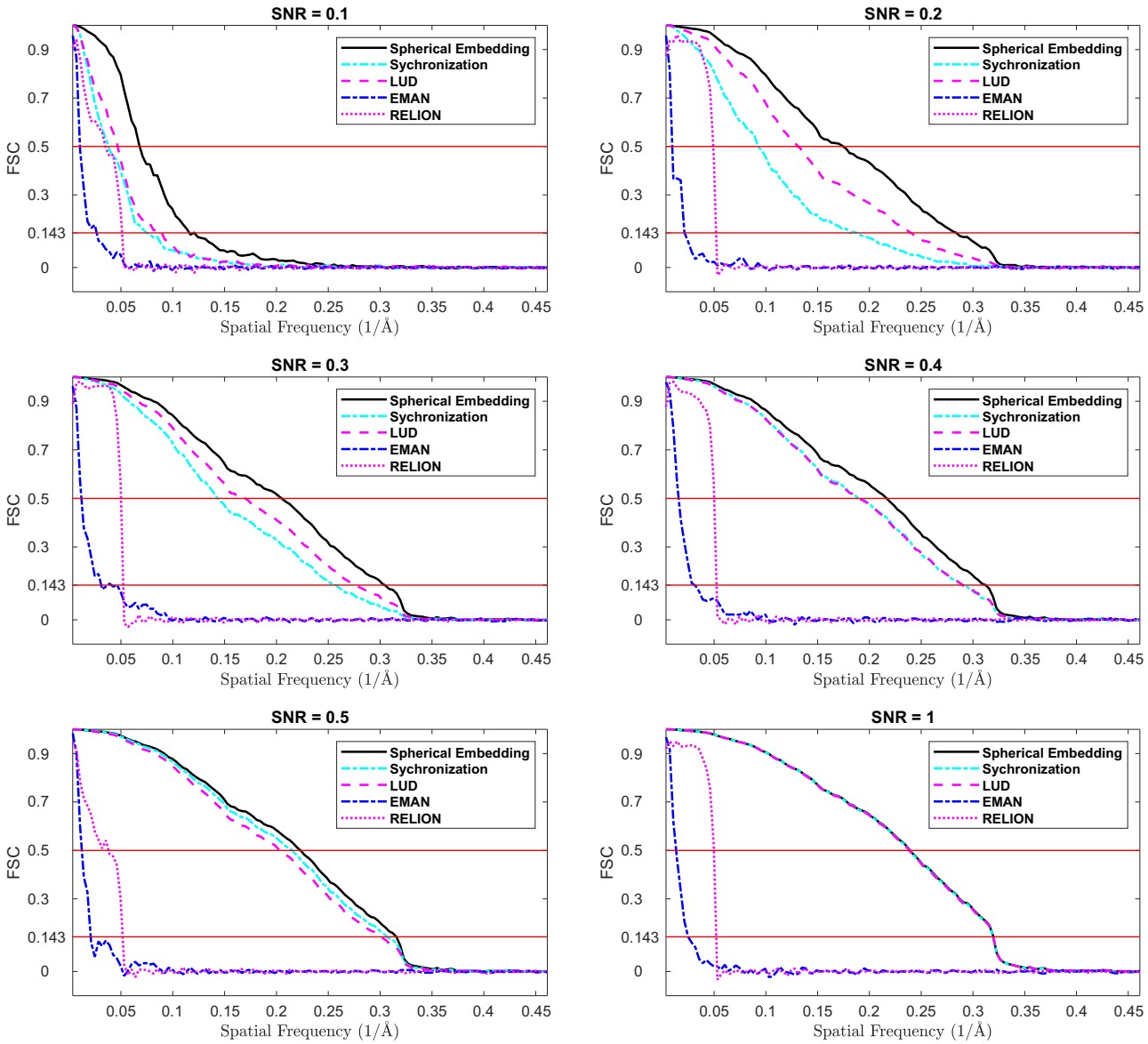

**Fig. 3** Comparison of the FSC curves produced at different SNR levels (0.1, 0.2, 0.3, 0.4, 0.5, and 1.0), by the proposed method (Spherical Embedding), Synchronization, LUD, EMAN 2.1, and RELION-2.

Synchronization, LUD all produced very similar maps with the reference map, not only fitting the overall shape but also representing some structural details. EMAN 2.1 produced a map with a roughly similar outline comparing with the reference map, while the map generated by RELION-2 lost much information as comparing with others especially at the 40S part. These results are consistent with the FSC curves shown in Fig. 5a.

The reason why EMAN 2.1 and RELION-2 do not perform well may be that only the 531 class averages are used in the reconstruction, without using any information from the original 11,983 images. The strategy is just set for a fair comparison, and not for exploring the full extent of the methods' capabilities.

**Evaluation of the 3D reconstructions for the EMPIAR-10328 dataset.** The second real projection data, particles of the Hedgehog receptor Patched (PTCH1) in complex with a conformation selective nanobody TI23, are downloaded from the EMPIAR database (EMPIAR-10328, ref. [31]). The corresponding EM density map EMD-22689 is downloaded as the reference map

for resolution estimation. The sampling rate for the dataset of EMPIAR-10328 is 1.059 Å/pixel. RELION-2 is used to produce 2D class averages from the original 307,652 particle images using the CTF parameters contained in the original star file. After 2D classifications are performed 5 times, manually selected 390 class averages with good quality and low similarity are produced, which is named ImageSet_D.

The ImageSet_D dataset is used as input to generate 3D initial models by Spherical Embedding, Synchronization, LUD, EMAN 2.1, and RELION-2, instead of using the original 307,652 particle images. For Spherical Embedding, Synchronization, and LUD, the same center alignment algorithm used for ImageSet_C is applied. For both EMAN 2.1 and RELION-2, the number of initial models is set to 1 for a fair comparison with the other three methods.

Resolutions are estimated by calculating FSC curves between the reconstructed density maps and the reference map EMD-22689. The FSC curves are shown in Fig. 6a, b. It is found that the FSC curves produced by Spherical Embedding, Synchronization, and

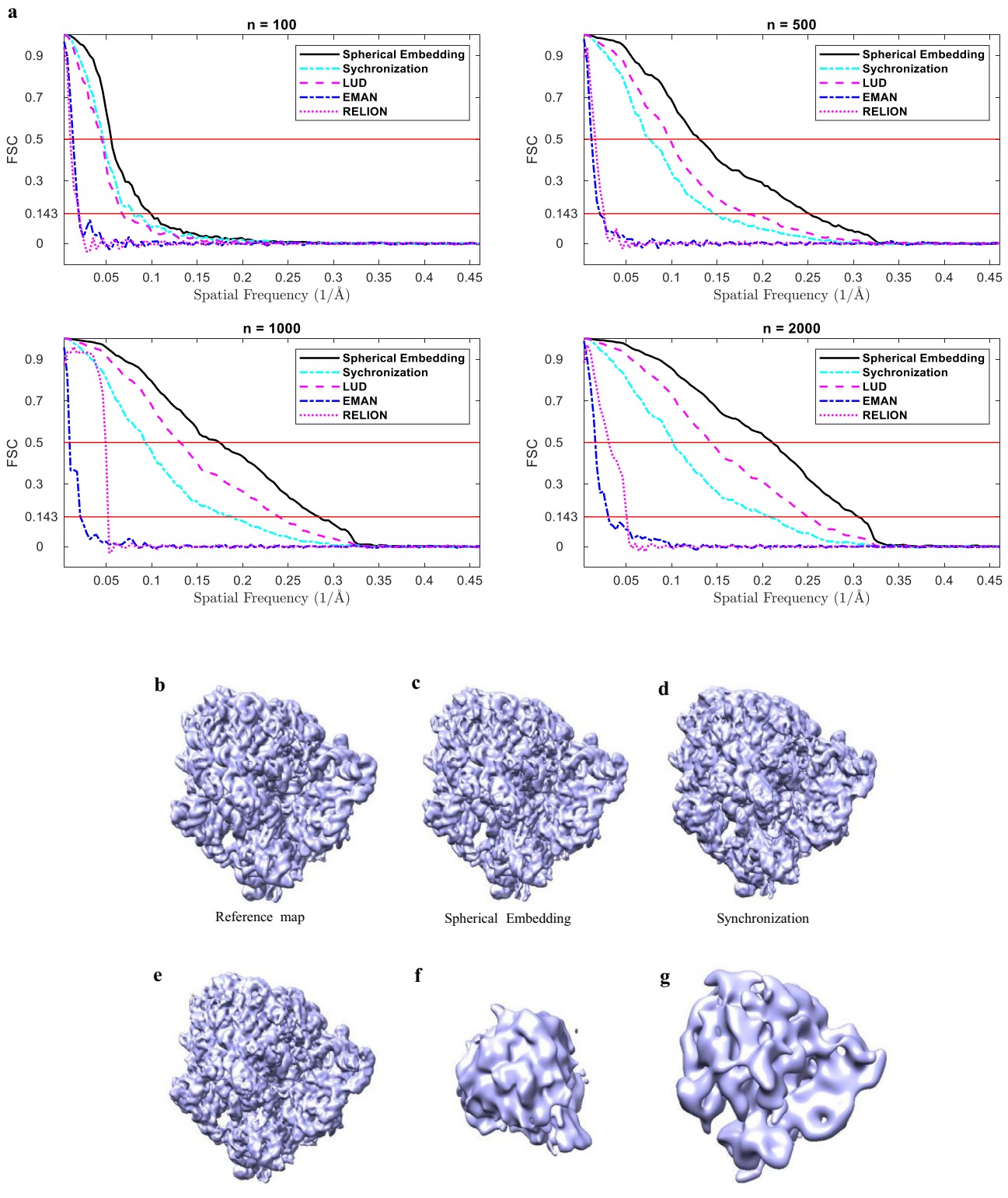

**Fig. 4 Comparison of the FSC curves produced from different number of projection images ($n = 100$, $n = 500$, $n = 1000$, and $n = 2000$), and the comparison of the reconstructions by different methods for the simulated data ($n = 1000$, SNR = 0.2). a** The FSC curves produced from different number of projection images ($n = 100$, $n = 500$, $n = 1000$, and $n = 2000$) by the proposed method (Spherical Embedding), Synchronization, LUD, EMAN 2.1, and RELION-2. **b** The reference density map (EMD-3508). Others are the reconstructed maps by: **c** the proposed method (Spherical Embedding), **d** Synchronization, **e** LUD, **f** EMAN 2.1, and **g** RELION-2.

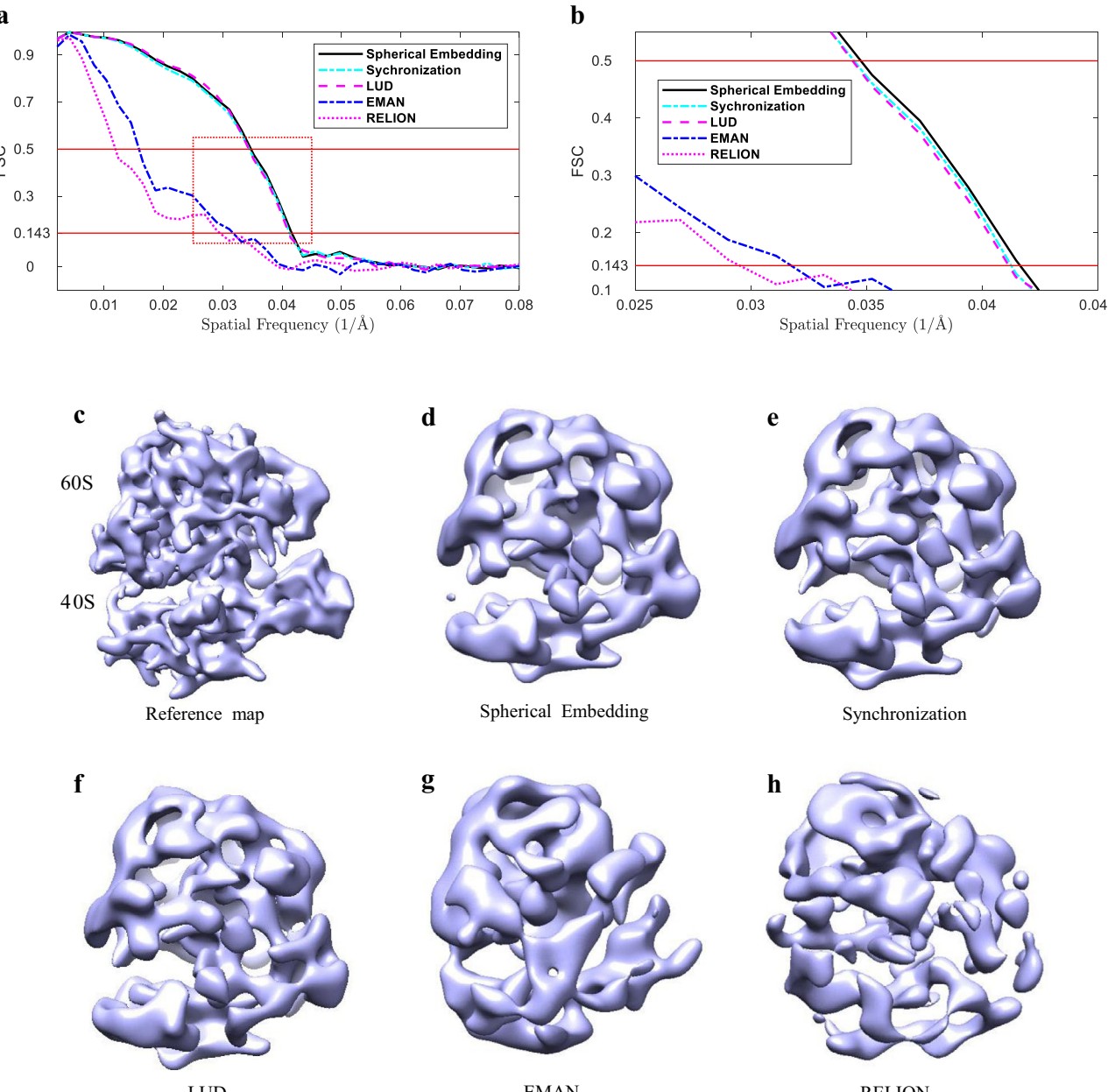

**Fig. 5 Evaluation using the FSC curves and the reconstructions for the EMPIAR-10028 dataset. a** Comparison of the FSC curves produced by the proposed method (Spherical Embedding), Synchronization, LUD, EMAN 2.1, and RELION-2. **b** The zoom-in of the red dashed rectangle region in (**a**), which shows the details of the FSC curves around the spatial frequency of 0.04 1/Å. **c** The reference density map (EMD-2660). Others are the reconstructed maps from the class averages by: **d** the proposed method (Spherical Embedding), **e** Synchronization, **f** LUD, **g** EMAN 2.1, and **h** RELION-2.

LUD are similar to each other, and they are all better than the curves produced by EMAN 2.1 and RELION-2 at FSC = 0.143. But in the low-frequency region, RELION-2 performs better than the other methods. From Fig. 6b which shows the details within the dashed red rectangle in Fig. 6a, it can be seen that the resolution of the map produced by Spherical Embedding is slightly better than both Synchronization and LUD. Similar to the ImageSet_C dataset, the performance gain by the Spherical Embedding method on this dataset is not as remarkable as on ImageSet_A and ImageSet_B, which may also be caused by the high SNR and the small number of the class averages in the ImageSet_D dataset.

Figure 6c through Fig. 6h show the reference map (EMD-22689) and the reconstructions by the five methods. The reconstructions of Spherical Embedding and Synchronization are similar. Compared

with Spherical Embedding and Synchronization, the reconstruction of LUD has less information in detail. Although the reconstructions of EMAN 2.1 and RELION-2 are similar with the reference map in the outline, much detailed information is lost compared to the other three methods. Similar to ImageSet_C, the reason why EMAN 2.1 and RELION-2 do not perform well on this real dataset may be that only the 390 class averages are used as input in the reconstruction, but not the original 307,652 images.

## Discussion
In this work, a new 3D initial model reconstruction method is proposed for SPA. Two spherical embeddings are used to determine the normal direction and the in-plane rotation of a projection image, respectively, which produces the projection angle

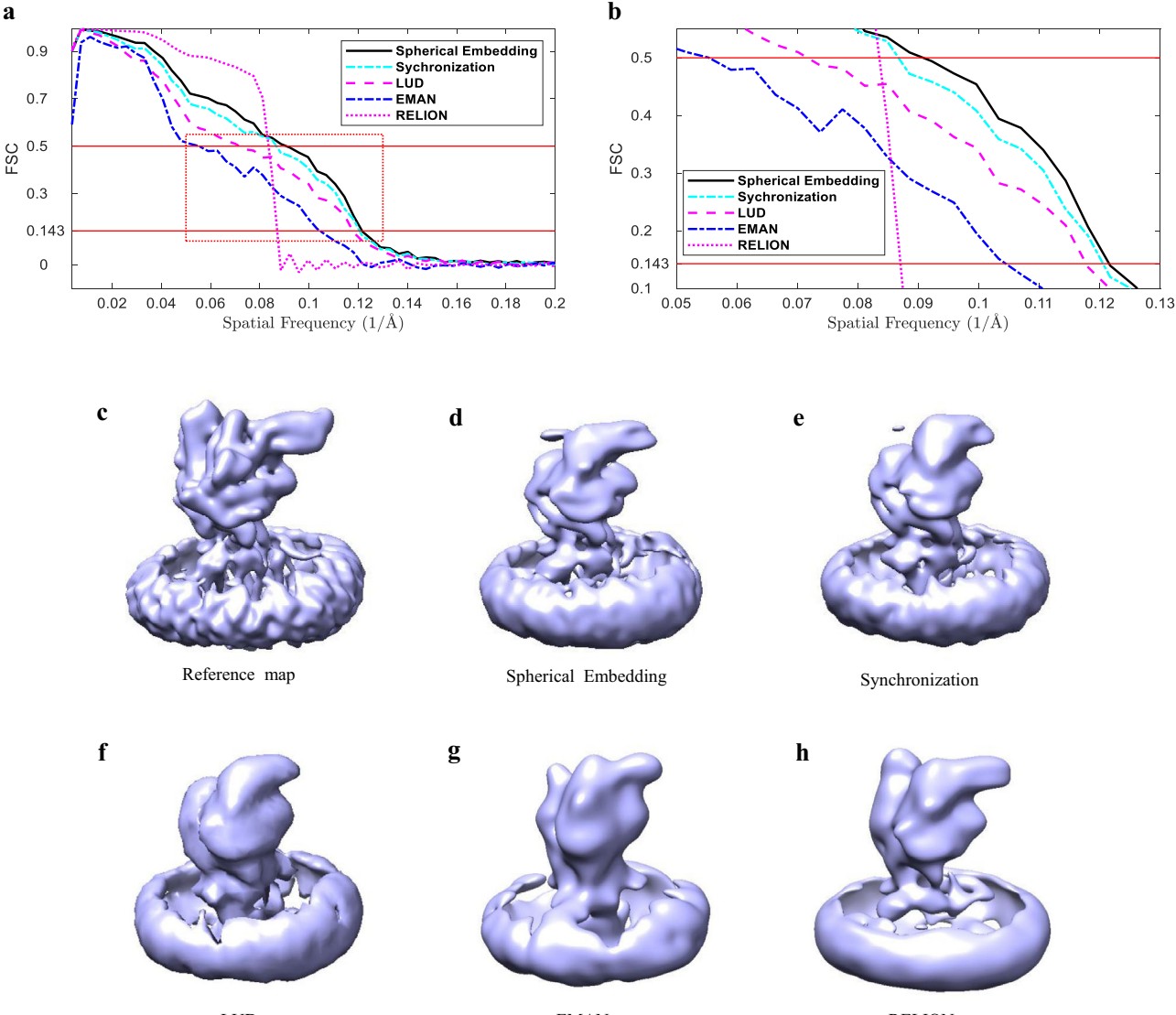

**Fig. 6 Evaluation using the FSC curves and the reconstructions for the EMPIAR-10328 dataset. a** Comparison of the FSC curves produced by the proposed method (Spherical Embedding), Synchronization, LUD, EMAN 2.1, and RELION-2 on the EMPIAR-10328 dataset. **b** The zoom-in of the red dashed rectangle region in (**a**), which shows the details of the FSC curves around the spatial frequency of 0.09 1/Å. **c** The reference map (EMD-22689). Others are the reconstructed maps from the class averages by: **d** the proposed method (Spherical Embedding), **e** Synchronization, **f** LUD, **g** EMAN 2.1, and **h** RELION-2. The density around the transmembrane region in the lower part of the initial models of the PTCH1::T123 complex belongs to the detergent micelle.

estimation by satisfying global consistency constraints derived from all the projection images. The proposed method can produce favorable results compared to other popular 3D reconstruction methods on both simulated datasets and two real datasets. The results show that the spherical embedding can be used to improve the initial model reconstruction for SPA. Since the result of the spherical embedding is directly related to the orientation of the projection image, the proposed method has the advantage of exploring the global consistency constraints directly and efficiently on the 3D spherical surface, which may contribute to its success of producing more accurate projection angle estimations than the other methods.

Although the proposed method can improve the 3D reconstruction result significantly when the number of the projection images is larger than 500 and the SNR level is as low as 0.2 or 0.1, it can only produce similar results with Synchronization and LUD in some other situations, such as smaller dataset or higher SNR levels. When applied to the real dataset, the performance gain of

the proposed method is not as remarkable as well compared to the other methods. Thus, it will be marked as the main goal of our future work to further improve the performance of the 3D reconstruction method for the real dataset.

## Methods

In the proposed method, the orientation of a projection image is represented by its normal vector and its local X-axis vector, which is equivalent to the representation by Euler angles $\alpha$, $\beta$, and $\gamma$ defined by van Heel et al.[20], as shown in Fig. 1. The Euler angles $\alpha$ and $\beta$ can be determined by the normal vector alone, while the rotation of the local X-axis vector along the normal vector gives the Euler angle $\gamma$. So, after the determination of the normal vector and local X-axis vector for each projection image, the orientation represented as Euler angles can be easily derived. In the paper, Euler angles $\alpha$ and $\beta$ defined here are also used to represent the position of a point on the unit spherical surface.

The main procedure of the proposed spherical embedding method is shown in Fig. 7. The method includes mainly 6 steps: (1) common line detection; (2) dihedral angle estimation; (3) the first 3D spherical embedding; (4) estimating angles between local X-axis vectors; (5) the second 3D spherical embedding; (6) alignment of the normal vectors and the local X-axis vectors.

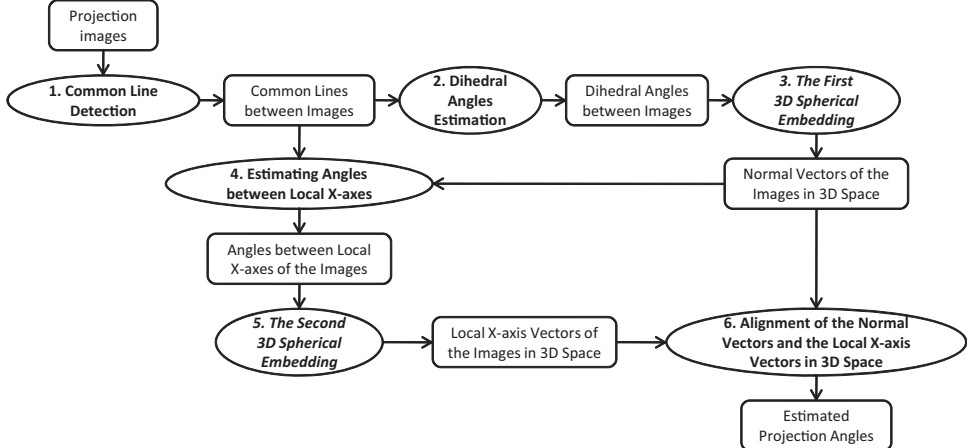

**Fig. 7** The flowchart of the proposed method. The ovals represent processing methods, and the rounded rectangles represent data.

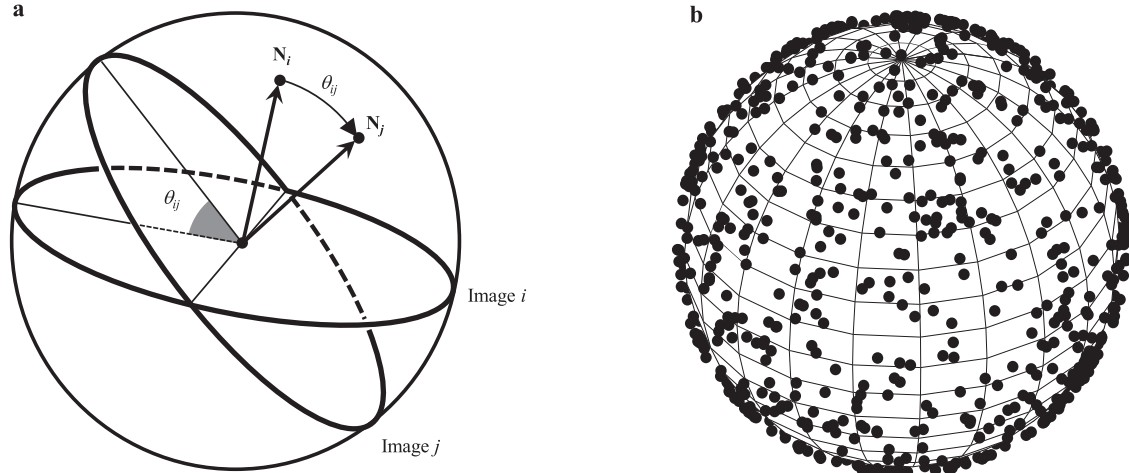

**Fig. 8 Illustration of the normal vectors and their intersection points on a unit spherical surface. a** The angle between the normal vectors of two images equals to their dihedral angle in the 3D reconstruction space. $N_i$ and $N_j$ are the intersection points of the normal vectors of image $i$ and image $j$, respectively, on the unit spherical surface. The distance between $N_i$ and $N_j$ on the unit spherical surface equals to the dihedral angle $\theta_{ij}$. **b** The spherical embedding of 1000 intersection points of the normal vectors on a unit spherical surface.

In ideal cases, based on the Central Section Theorem[21], the 2D Fourier transform of a projection image is the same as the slice perpendicular to the projection angle in the 3D Fourier transform of the 3D protein model. So, two projection images with different orientations will intersect at a common line in the Fourier space. In step (1), the common line between two projection images can be detected using normalized cross correlation[20].

Using the common line information, the dihedral angles between all pairs of images can be estimated by the voting method introduced by Singer et al.[22]. For a pair of images $i$ and $j$, using a different third image $k$ usually gives a different dihedral angle $\theta_{ij}^{(k)}$ between the image pair due to the errors introduced in the common line estimation. The voting method will consider the dihedral angles computed using all the other images $k \neq i, j$, and select the most popular angle estimation by creating a histogram of the computed dihedral angles $\theta_{ij}^{(k)}$:

$$hist_{ij}(t) = \sum_{k \neq i,j} \frac{1}{\sqrt{2\pi\sigma^2}} e^{-(\sigma t - \theta_{ij}^{(k)})^2/(2\sigma^2)}, \; \sigma = \pi/T, \; t \in [0, ..., T-1] \quad (1)$$

The final estimation of the dihedral angle $\theta_{ij}$ corresponds to the maximum height of the histogram, $h_{ij}$, which indicates the reliability of the estimation:

$$t_{max} = \operatorname*{arg\,max}_{t \in [0, ..., T-1]} \{hist_{ij}(t)\} \quad (2)$$

$$h_{ij} = hist_{ij}(t_{max}) \quad (3)$$

$$\theta_{ij} = t_{max}\sigma \quad (4)$$

Using the voting method, the dihedral angles between all pairs of images in step (2) can be estimated, which is more reliable compared to the original common line information.

For any pair of projection images, the dihedral angle between them in the 3D reconstruction space equals to the angle between their normal vectors. After the dihedral angles between all pairs of images are determined, they are used as the initial values for the angles between the corresponding normal vectors. If the normal vectors are at unit length, the angles between them can be treated as the distances between the intersection points of the normal vectors on a unit spherical surface as shown in Fig. 8a.

In the ideal case, all the dihedral angles are consistent to each other. According to the Central Section Theorem, all the projection images in the Fourier space pass through the same center point, so their normal vectors should pass through the same center point too, and all the intersection points of the normal vectors can be embedded on the same 3D unit spherical surface together.

But in reality, these estimated dihedral angles are usually not consistent to each other due to the errors introduced during the early computation. Suppose that the distances between the intersection points of the normal vectors on the unit spherical surface all satisfy the triangular inequality, these intersection points can be treated as the objects embedded in a metric space. For 3 intersection points, they can be embedded on a 3-dimensional spherical surface together while preserving the distance values. For $n$ (>3) intersection points, to preserve all the distances between intersection points, they can be embedded on a $n$-dimensional spherical surface, but they usually cannot be embedded on a 3-dimensional spherical surface if there are inconsistencies caused by errors in the earlier computations. To produce an embedding of all the $n$ intersection points on a 3-dimensional spherical surface as shown in Fig. 8b, some initial distances between the intersection points have to be modified, which can be realized by the spherical embedding method[26].

After the spherical embedding, a consistent solution to the normal directions of all the projection images is found. The spherical embedding not only can satisfy the

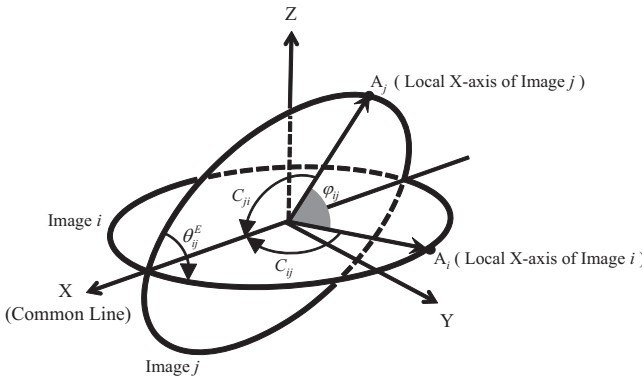

**Fig. 9 Computation of the angle between the local X-axis vectors of image $i$ and image $j$.** Image $i$ is on the XY plane. The common line between image $i$ and image $j$ is along X-axis.

consistency constraints of all the dihedral angles, but also can rectify the errors introduced in the earlier steps of the projection angle estimation by modifying the initial distances between the intersection points. This is the key observation which leads to the design of the proposed method.

In the step (3) of the proposed method, the 3D spherical embedding method proposed by Wilson et al.[27] is used to estimate the intersection points of the normal vectors, using the dihedral angles $\theta_{ij}$ computed in step (2) as initial distance values. The goal of the 3D spherical embedding is to find an embedding of the intersection points with Euler angles $(\alpha_i, \beta_i)$, $i = (1, 2, \ldots, n)$, on the 3D unit spherical surface, such that the distortion of the distances, $\varepsilon$, defined as follows is minimized:

$$\varepsilon = \sum \omega_{ij}(\theta_{ij} - \theta_{ij}^E)^2 \qquad (5)$$

where $\theta_{ij}^E = arc\cos(\cos\beta_i\cos\beta_j + \sin\beta_i\sin\beta_j\cos(\alpha_i - \alpha_j))$ is the dihedral angles after spherical embedding, and $\omega_{ij}$ is the weight which is determined by the maximum height of the histogram computed in the voting method in step (2) in the proposed method:

$$\omega_{ij} = \begin{cases} h_{ij} & if\ h_{ij} > H \\ 0 & otherwise \end{cases} \qquad (6)$$

where $H$ is the $10th$ percentile value from the set $\{h_{ij}\,|\,i = 1, 2, \ldots, n; j = 1, 2, \ldots, n\}$.

Since there may be two mirror rotations that both satisfy the distance constraints, to select the correct one, the common line between a good pair of projection images which have the highest $h_{ij}$ score produced by the voting method can be used. The correct rotation will rotate the common lines on both projection images to a similar direction in the 3D space.

Until now, the embedding results give the two Euler angles $(\alpha_i, \beta_i)$, $i = (1, 2, \ldots, n)$, which determine the normal vectors of all the projection images. The rest steps of the method is used to find another Euler angle $\gamma_i$, $i = (1, 2, \ldots, n)$.

To estimate the initial value of the angles between local X-axis vectors in step (4), the common line information as well as the dihedral angles after the spherical embedding are used. As shown in Fig. 9, for any two images $i$ and $j$, a coordinate system can be setup such that image $i$ is on the XY plane, and the common line between the two images is along the X-axis. If $C_{ij}$ represents the angle between the common line and the local X-axis vector of the image $i$, $C_{ji}$ represents the angle between the common line and the local X-axis vector of the image $j$, $\theta_{ij}^E$ is the dihedral angles between the two images after the first spherical embedding, $A_i$ is the intersection point of the local X-axis vector of image $i$ on the unit spherical surface, and $A_j$ is the intersection point of the local X-axis vector of image $j$ on the unit spherical surface, it can be seen that the coordinate of $A_i$ is $(\cos C_{ij}, \sin C_{ij}, 0)$, the coordinate of $A_j$ is $(\cos C_{ji}, \sin C_{ji}\cos\theta_{ij}^E, \sin C_{ji}\sin\theta_{ij}^E)$, and the angle between the two local X-axis vectors is:

$$\varphi_{ij} = arc\cos(\cos C_{ij}\cos C_{ji} + \sin C_{ij}\sin C_{ji}\cos\theta_{ij}^E) \qquad (7)$$

Similarly, in step (5), the 3D spherical embedding of the intersection points of the local X-axis vectors on the unit spherical surface is carried out using the method proposed by Wilson et al.[26], where the angles $\varphi_{ij}$ computed in step (4) are used as initial values and $\omega_{ij}$ as the weights. The resulting embedding with Euler angles $(\alpha'_i, \beta'_i)$, $i = (1, 2, \ldots, n)$, minimizes the error:

$$\varepsilon'' = \sum_{i \neq j} \omega_{ij}(\varphi_{ij} - \varphi_{ij}^E)^2 \qquad (8)$$

where $\varphi_{ij}^E = arc\cos(\cos\beta'_i\cos\beta'_j + \sin\beta'_i\sin\beta'_j\cos(\alpha'_i - \alpha'_j))$.

Finally, in step (6), the two spherical embedding results are aligned using an orthogonal constraint that the local X-axis vector should be perpendicular to the

normal vector for each image. This is done by finding a rotation matrix $R$:

$$R = \begin{bmatrix} R_{11} & R_{12} & R_{13} \\ R_{21} & R_{22} & R_{23} \\ R_{31} & R_{32} & R_{33} \end{bmatrix} \qquad (9)$$

which minimizes the following error that is the summation of all the dot products of the local X-axis vectors and the corresponding normal vectors:

$$\varepsilon'' = \sum_{i=1}^{n} \begin{bmatrix} \sin\beta''_i\cos\alpha''_i \\ \sin\beta''_i\sin\alpha''_i \\ \cos\beta''_i \end{bmatrix}^T \cdot \begin{bmatrix} \sin\beta_i\cos\alpha_i \\ \sin\beta_i\sin\alpha_i \\ \cos\beta_i \end{bmatrix} \qquad (10)$$

$$= \sum_{i=1}^{n} [\cos\beta_i\cos\beta''_i + \sin\beta_i\sin\beta''_i\cos(\alpha_i - \alpha''_i)]$$

where $\begin{bmatrix} \sin\beta''_i\cos\alpha''_i \\ \sin\beta''_i\sin\alpha''_i \\ \cos\beta''_i \end{bmatrix} = \begin{bmatrix} R_{11} & R_{12} & R_{13} \\ R_{21} & R_{22} & R_{23} \\ R_{31} & R_{32} & R_{33} \end{bmatrix} \cdot \begin{bmatrix} \sin\beta'_i\cos\alpha'_i \\ \sin\beta'_i\sin\alpha'_i \\ \cos\beta'_i \end{bmatrix}$

Because there are 9 unknown variables in the rotation matrix $R$ and the image number $n$ is usually much larger than 9, the problem can be transformed into an overdetermined homogenous linear system:

$$\begin{bmatrix} \sin\beta''_i\cos\alpha''_i \\ \sin\beta''_i\sin\alpha''_i \\ \cos\beta''_i \end{bmatrix}^T \cdot \begin{bmatrix} \sin\beta_i\cos\alpha_i \\ \sin\beta_i\sin\alpha_i \\ \cos\beta_i \end{bmatrix}$$

$$= \begin{bmatrix} \sin\beta'_i\cos\alpha'_i \\ \sin\beta'_i\sin\alpha'_i \\ \cos\beta'_i \end{bmatrix}^T \cdot \begin{bmatrix} R_{11} & R_{12} & R_{13} \\ R_{21} & R_{22} & R_{23} \\ R_{31} & R_{32} & R_{33} \end{bmatrix}^T \cdot \begin{bmatrix} \sin\beta_i\cos\alpha_i \\ \sin\beta_i\sin\alpha_i \\ \cos\beta_i \end{bmatrix} = \begin{bmatrix} 0 \\ 0 \\ 0 \end{bmatrix} \qquad (11)$$

where $i = 1, 2, \ldots, n$, and $R_{kl}$ ($1 \leq k \leq 3, 1 \leq l \leq 3$) are the 9 unknown variables.

In the proposed method, the least square method is used to solve the overdetermined system. For $Ax = 0$, where $x$ represents $R_{kl}$ ($1 \leq k \leq 3, 1 \leq l \leq 3$), the approximate solution of the system is the eigenvector corresponding to the smallest eigenvalue of the matrix $A^T A$.

After the rotation $R$ is found, the Euler angles $(\alpha''_i, \beta''_i)$, $i = (1, 2, \ldots, n)$ which represent positions of the aligned X-axes can be obtained. Then the Euler angle $\gamma_i$, $i = (1, 2, \ldots, n)$, can be determined using the following equation:

$$\begin{bmatrix} \cos\gamma_i \\ \sin\gamma_i \\ \delta_i \end{bmatrix} = \begin{bmatrix} \cos\beta_i & 0 & -\sin\beta_i \\ 0 & 1 & 0 \\ \sin\beta_i & 0 & \cos\beta_i \end{bmatrix} \cdot \begin{bmatrix} \cos\alpha_i & \sin\alpha_i & 0 \\ -\sin\alpha_i & \cos\alpha_i & 0 \\ 0 & 0 & 1 \end{bmatrix} \cdot \begin{bmatrix} \sin\beta''_i\cos\alpha''_i \\ \sin\beta''_i\sin\alpha''_i \\ \cos\beta''_i \end{bmatrix} \qquad (12)$$

In the ideal case, $\delta_i$ in the above equation is zero. So, the value of $\sum_{i=1}^{n} \delta_i^2/n$ can be used as an indication of the consistency of the two spherical embeddings, with a smaller value indicates a better result.

After the projection angles are estimated, the FIRM method included in the package ASPIRE 0.14 (http://spr.math.princeton.edu) can be used to produce the final 3D reconstruction result.

**Statistics and reproducibility**. The experiments were repeated several times, and yielding similar results. When generating the simulated datasets, Gaussian noises with different random seeds were used in the experiments. Researchers were blinded to the expected results. Different researchers were responsible for data preparation, coding, result collection, and data analysis.

**Reporting summary**. Further information on research design is available in the Nature Research Reporting Summary linked to this article.

## Data availability

All the data needed for repeating the experiments described in the paper are available at https://github.com/yluATlzu/3DReconstruction_SE. We have used the following publicly available datasets: EMD-3508 (cryo-EM structure of Escherichia coli 70S ribosome-ArfA-RF2 complex, https://www.emdataresource.org/EMD-3508), EMD-2660 (cryo-EM structure of the Plasmodium falciparum 80S ribosome, https://www.emdataresource.org/EMD-2660), EMPIAR-10028 (micrographs and particle coordinates of a Plasmodium falciparum 80S ribosome, https://www.ebi.ac.uk/empiar/EMPIAR-10028/), and EMPIAR-10328 (micrographs and particle coordinates of PTCH1-TI23 complex, https://www.ebi.ac.uk/empiar/EMPIAR-10328/).

## Code availability

The code used in the paper for 3D reconstruction using spherical embedding is available at https://github.com/yluATlzu/3DReconstruction_SE. Before running the 3D reconstruction code, ASPIRE 0.14, which is available at http://spr.math.princeton.edu, needs to be installed first.

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

## Acknowledgements

This work is supported by the National Key R&D Program of China (Grants Nos. 2017YFE0111900, 2018YFB1003205).

## Author contributions

Y.L. designed the 3D reconstruction method using spherical embedding, developed the code, and wrote the manuscript. J.L. and B.Z. both contributed to the code development and performed the experiments. L.Z. selected and prepared the datasets, performed the experiments using RELION and EMAN, and revised the manuscript. J.H. contributed to the experiment design and the manuscript writing.

## Competing interests

The authors declare no competing interests.
