## [Peer Review File · Communications Biology]

Reviewers' comments:

Reviewer #1 (Remarks to the Author):

Remarks to the author

The manuscript by Y.Lu and co-authors "3D Reconstruction from Cryo-EM Projection Images Using Two Spherical Embeddings" reports a method that can be used for refinement of angular orientations of particle images (projections). This is a rather important step that precedes a structural reconstruction of 3D densities using cryo-EM images of the protein complexes.

From the mathematical point of view the reported approach represents a kind of modification of the angular reconstitution which has a potential for broader usage of this method. However, while the main ideas were published in 1987(doi: 10.1016/0304-3991(87)90078-7) and 1995 (DOI: 10.1038/nsb0195-18), the method did not attained a broad recognition due to closed codes and limited access to the software. In this paper the authors tried to develop further the ideas implemented while ago and to write a new package available for an average user. This is a brilliant idea. However, the materials delivered in this paper should be made more graspable for an average user, who does not have an advanced knowledge in geometry (in math more generally). Part with methods should go before the section with results. Results should be more convincing. The authors write "...the estimated common lines are usually inconsistent among different image pairs, which cannot be satisfied together during the 3D reconstruction..." Yes this is the point of this publication. It was important to do such analysis reported in this paper. However, the inconsistency in angles due to the lower SNR issues cannot be resolved by the reconstruction calculation in principle. That has to be done at the step if determination of angels in advance, before the reconstruction calculations otherwise results will be unsatisfactory, figure 6 confirm that.

Unfortunately the authors provided very little information on how they will do that while using rather wordy, lengthy referring to the other papers and specific, not usual, terminology. The language of the paper submitted is rather specific. The authors were trying to invent their own language (terminology) that is not quite common for the users in electron microscopy, who are supposed to be the main users of this new package. The authors (although they are citing other publications where some terms were used) should explain the terms used and use then the abbreviation accepted in cryo-EM field (see below).

It was rather surprising that the authors apparently were not able to make the initial determination of orientations (even the not quite correct ones) by their own software, because they have started from the angles obtained by another package. They were writing that the "Voting method" was not very good, but they have used as the initial angles the orientations obtained by this software. Why they did not use just random assignment of the angles for the projections? What is a specificity of the Voting method and it has been used for the initial determination of particle image orientations? It seems that this packaged developed by the authors has been exclusively developed for the refinement of angles.

To determine only dihedral angles between projections is not enough to define the orientations of the projections. To calculate the reconstruction we need to know all three Euler angles. Moreover, the angles have to be brought to the common system of coordinates related to the reconstruction that is going to be calculated. The authors did not explained how three Euler's angles alpha (rotation in a plain of the image), beta (the tilt of a projection from the Z axis of the object), and gamma (rotation of a view (projection) around of the Z axis of the object) are related to the dihedral angles between projections. They did not explain how the overall coordinate system has to be defined.

The authors did not show their reference maps and reconstructions determined before and after refinement of angles. It was strange appearance of the FS curves... The graphs are not consistent between different packages. So far Eman and Relion were good and providing very reasonable results. The authors did no show results of the model reconstructions In this paper the authors wrote that spherical embedding was not doing calculations of 3D map, that it was done in different software. How the reference map was obtained? Apparently the authors have used a very strange

ban pass filtering of the original map and have distorted quite important features of the structure. The overall resolution of the map was about 25 Angstrom, the other reconstructions indicate definitely some problems with determination of angles used in the reconstructions. Apparently the authors did not realize, that nowadays in EM structural studies scientists do not use anymore classes of 2D particle images, only raw single images of particles without any averaging. That makes determination of orientations on single images significantly more challenging. The authors make too many conjectures in their report and reconstructions.

Recommendations:

Please replace SPR with SPA (Single Particle Analysis (not reconstruction)), which is now a broadly excepted term.

Typically in EM the word "symmetricity" is not used. The standard term is "symmetry". A protein complex may have some symmetry that typically is described by one of the point groups of symmetry.

The concept of the "embedding space" was taken from one of the cited papers, but was not explained. It is not a "natural" space. Please explain to a reader in simple terms: what is a difference between a Spherical Embedding and the Euler sphere. How can be "embedding", which is a concept at least in 3D space, can be done on 2D surface (sphere)? This illogical link has to be explained.

Please explain to a reader in simple terms: why one need to have two Spherical Embedding instead of one Euler sphere? In your figure 8b no coordinates are indicated. What are they? Please specify designation of all angles used figures 1 and 9, they were shown different, but in reality they are the same making the reader completely confused.

Please do numbering of your formulas.

Please explain in which system of coordinates you are looking for projection orientation? It would be a tedious job and extremely time consuming to analyse all possible combinations of projections pairs if the system coordinates is not fixed (not bound to a certain orientation of the object in space). How the task of alignment of these multiple dihedral angles (one projection N with the other 1000-1 projections and the next N+1 with the other 1000-1 projections will be aligned in space and what you will do with 1000 such spherical distributions in space? They must be adjusted somehow? Shouldn't they? What sort of information can you get out of such alignments? How is it related to the Euler orientation of each projection?

Please explain how the orientation of a projection image (three coordinates of one projection: alpha, beta, gamma) can be represented by two points on the spherical surface, and the global consistency constraints? Where the two points are coming from for one projection? Where are they located and what it a mathematical link between them?

"...Taken together, the spherical embedding in the proposed method is very effective for producing accurate estimations of the dihedral angles, which correspond to the Euler angles alpha and beta...". The authors did not explained clearly how dihedral angles between three projections can provide two Euler angles; the corresponding figure should be presented in the paper. The noise problem is rather obvious, so a general statistic should be taken in account. However, the method part was not explained it clearly.

Reviewer #2 (Remarks to the Author):

This is a fine and interesting manuscript. It leverages the understanding of the geometry of projections to utilize algorithms involving data on the sphere (Spherical Embedding). The performance of the algorithm seems to be a significant improvement in assigning angles to projections over other programs (EMAN, relion).

I have some minor grammatical criticisms. I would replace symmetricity with symmetry, where it appears.

Section 2.1.1

"which are called as the " -> "which are called the".

Reviewer #3 (Remarks to the Author):

The manuscript of Lu and colleagues offers a refined approach on an important theoretical aspect in cryo-electron microscopy (cryo-EM): calculating the initial orientation of 2D images for reconstructing a 3D object. While the mathematical problem is rather direct, the low signal to noise signal present in cryo-EM images imposes serious challenges for algorithms trying to find a numerical solution. With the assumption that all the particles present in the sample are identical, following the well known Fourier slice theorem, the Fourier transforms of these 2D images became slices through the Fourier transform of the object to be reconstructed. This means that any two images will share a 'common line' in Fourier space. The algorithms used in cryo-EM reconstruction use different strategies to find common lines in order to reconstruct the 3D model. However, the initial estimation of these angles is crucial for the success of the whole enterprise. While different algorithms take great care that the refinement will finally converge to the correct solution, there is always the possibility that the iterations will reach a local minimum and therefore provide a wrong answer.

The present manuscript proposes a solution to avoid the entrapment in a local minimum and to correctly estimate the initial orientation of each single 2D image. The idea is to accurately find the orientation of one image by calculating the common line between this image and all the other images. For this, it is convenient to replace the Euler angles as the parameters which describe the direction of the projection with an equivalent description based on two vectors: the normal vector and a local X-axis vector. This allows to define the position of the common line between two images based on the dihedral angle between two images as dictated by their two normal vectors. The method implies a 'voting' system in which the angle between the i-image and the j-image is estimated using the histogram between all the dihedral angles calculated using all the remaining images. The correct answer is the maximum value of this histogram. In an ideal case, all these dihedral angles would be consistent to each other. However, the reality is more complex; and to deal with the errors present in real images, the authors use 'spherical embedding' – a method which satisfies all the constraints and can simultaneously rectify errors.

The authors evaluate their method using both simulated and real projection images. The method performs well in both cases surpassing the EMAN2 and Relion algorithms as well as another two related methods which use a 'voting system' for common line calculation.

Under the constant flow of impressive cryo-EM structures, it is salutary to see that attention is also devoted to enhancing the arsenal of algorithms used in the cryo-EM process.

The article is very well written, the results are clearly organised. The authors provide an exemplar description of the methodology: all the definitions are provided, the rationale of each step is clearly stated, and the experiments are clearly described. However, I have to say, the article is rather theoretical and the 'real images' test is limited to a ribosome data set. Without a richer palette of relevant biological examples of different sizes and complexity, the manuscript would be a better fit for a specific journal, such as *Journal of Structural Biology*.

Comments:

There is a major issue which I would like to be addressed by the authors. Finding the correct orientation of a common line between two images is strongly dependent on the centring of each particle. At no point in the manuscript do the authors indicate any concern regarding this point. For the simulated images, the authors do just rotate the map in order to generate the artificial projections already centered. For the 'real images' case, they use 2D averages and not original images – a reasonable way to deal with the signal to noise ratio. As they do not mention particle picking, I suppose they used directly the EMPIAR data which raises the question how accurate was the centring and how dependent is their method of the variations of this parameter.

Response to Referees

We would like to thank all the referees for their valuable comments which have helped us a lot in improving the manuscript. All the issues raised by the reviewers have been carefully considered and addressed. The wording and the organization of the paper are improved. We have made a lot of efforts to make the paper more understandable, by rewriting most of the Introduction section and other parts. More experimental results and discussions were added. The code and data used in the paper are also updated at https://github.com/yLuATLzu/3DReconstruction_SE. The main ideas of the paper are emphasized. Overall, we have revised the manuscript according to the comments of all the referees, where all modified parts are highlighted in the manuscript. In the following, our detailed responses (in blue color) to every concern raised by the referees are listed.

Response to Reviewer 1

We very much appreciate the comments and suggestions from Reviewer 1, which help to significantly improve the quality and description of the manuscript. We have revised the manuscript according to all the comments of Reviewer 1, where all changes have been highlighted in the manuscript. In the following text, the point-by-point replies are summarized:

1. The Reviewer commented:

The manuscript by Y.Lu and co-authors “3D Reconstruction from Cryo-EM Projection Images Using Two Spherical Embeddings” reports a method that can be used for refinement of angular orientations of particle images (projections). This is a rather important step that precedes a structural reconstruction of 3D densities using cryo-EM images of the protein complexes.

From the mathematical point of view the reported approach represents a kind of modification of the angular reconstitution which has a potential for broader usage of this method. However, while the main ideas were published in 1987(doi: 10.1016/0304-3991(87)90078-7) and 1995 (DOI: 10.1038/nsb0195-18), the method did not attain a broad recognition due to closed codes and limited access to the software. In this paper the authors tried to develop further the ideas implemented while ago and to write a new package available for an average user. This is a brilliant idea.

[Answer]: Thanks for the positive comments!

2. **The Reviewer commented:**

However, the materials delivered in this paper should be made more graspable for an average user, who does not have an advanced knowledge in geometry (in math more generally).

[Answer]: Thank you for the suggestion! The original description of the terms is not easy to be understood. Some newly introduced terms were not described clearly in the original manuscript. In order to make the paper more understandable, the article was revised a lot. For example, in the revised version, “normal vector”, “local X-axis vector” and “dihedral angle” are all introduced in a more understandable way.

3. **The Reviewer commented:**

The authors write “...the estimated common lines are usually inconsistent among different image pairs, which cannot be satisfied together during the 3D reconstruction...” Yes this is the point of this publication. It was important to do such analysis reported in this paper. However, the inconsistency in angles due to the lower SNR issues cannot be resolved by the reconstruction calculation in principle. That has to be done at the step of determination of angles in advance, before the reconstruction calculations otherwise results will be unsatisfactory, figure 6 confirm that.

[Answer]: Thank you for raising this point. And we apologize for the unclear description. The inconsistency problem of the common lines among different image pairs is solved after the voting method and before the 3D reconstruction in the proposed method using Spherical Embedding. The voting method can be used to calculate the common lines of the images, but cannot be used to

calculate the projection angles. The proposed method uses dihedral angles calculated from the common lines as input, and produces the consistent projection angles as output. Then the reconstruction is done using the FIRM method in Aspire 0.14, given the consistent projection angles as input. So, the problem of consistency is solved before the reconstruction, not during the reconstruction.

In the Introduction Section (Page 3, Paragraph 4), we have added the following sentence:

The voting method can be used to estimate the common lines of the projection images, but cannot be used to directly estimate the projection angles.

In the Introduction Section (Page 5, Paragraph 1), we have added the following sentences:

First, the dihedral angles between all the pairs of projection images are estimated from the common lines using the voting method. ... Using the spherical embedding, the intersection points of the normal vectors of the projection images on the 3D unit spherical surface can be determined, which gives the consistent dihedral angles in the 3D space. The consistent angles between X-axis vectors of the projection images in the 3D space can be produced similarly using another spherical embedding.

The last sentence in the Introduction Section (Page 6, Paragraph 2), “The benefit of using the spherical embedding compared with other methods for dealing with the inconsistency is that ..., leading to a more accurate and robust 3D reconstruction”, has been modified to:

The benefit of using the spherical embedding compared with other methods for dealing with the inconsistency is that ..., leading to a more accurate and robust projection angle estimation.

4. **The Reviewer commented:**

Unfortunately the authors provided very little information on how they will do that while using rather wordy, lengthy referring to the other papers and specific, not usual, terminology. The language of the paper submitted is rather specific. The authors were trying to invent their own language (terminology) that is not quite common for the users in electron microscopy, who are supposed to be the main users of this new package. The authors (although they are citing other publications where some terms were used) should explain the terms used and use then the abbreviation accepted in cryo-EM field (see below).

[Answer]: Thanks for your suggestion. The original expression of the article is easy to be misunderstood. This is mainly because some concepts were not introduced clearly in the original manuscript. In order to make the paper more understandable, the article was revised a lot in the Introduction section. Some terms have been renamed; other terms have been revised such that they are consistent in the whole article. For example, “end point” is renamed as “intersection point”, “X-axis vector” is replaced with “local X-axis vector”, the “orientation of the projection image” is used when talking about the concept while the “projection angle” is used as one of the representation of the orientation, *etc.* The corresponding terms are also explained when they first appear in the article.

5. **The Reviewer commented:**

It was rather surprising that the authors apparently were not able to make the initial determination of orientations (even the not quite correct ones) by their own software, because they have started from the angles obtained by another package. They were writing that the “Voting method” was not very good, but they have used as the initial angles the orientations obtained by this software. Why they did not

use just random assignment of the angles for the projections? What is a specificity of the Voting method and it has been used for the initial determination of particle image orientations? It seems that this packaged developed by the authors has been exclusively developed for the refinement of angles.

[Answer]: We apologize for the unclear description. The inputs to the proposed method are the dihedral angles calculated from the common lines. These dihedral angles are usually inconsistent with each other in the 3D space. Dihedral angle is only the angle between the normal vectors of two projection images, not the projection angle. So, what the proposed method do is to compute the consistent projection angles from the inconsistent dihedral angles (computed from the inconsistent common line information), which is not a refinement of the projection angles. Obviously, the input of the proposed method are the dihedral angles, and cannot be random assignment.

In the Introduction Section (Page 3, Paragraph 4), we have added this sentence:

The voting method can be used to estimate the common lines of the projection images, but cannot be used to directly estimate the projection angles.

In the Introduction Section (Page 5, Paragraph 1), we have added this sentence

First, the dihedral angles between all the pairs of projection images are estimated from the common lines using the voting method.

6. The Reviewer commented:

To determine only dihedral angles between projections is not enough to define the orientations of the projections. To calculate the reconstruction we need to know all three Euler angles. Moreover, the angles have to be brought to the common system of coordinates related to the reconstruction that is going to be calculated.

The authors did not explained how three Euler's angles alpha (rotation in a plain of the image), beta (the tilt of a projection from the Z axis of the object), and gamma (rotation of a view (projection) around of the Z axis of the object) are related to the dihedral angles between projections. They did not explain how the overall coordinate system has to be defined.

[Answer]: We apologize for the unclear description. In the proposed method, the dihedral angles and the angles between local X-axis vectors are used to represent the relative orientations between two projection images. The dihedral angle is only related to the Euler angles α and β , while the local X-axis vectors are needed for determining Euler angle γ (the in-plane rotation).

In the Introduction Section (Page 4, Paragraph 2), we have added the following to explain the relationship between our representation (normal vector and local X-axis vector) of the orientation and the Euler angles:

In our method, the orientation of a projection image is represented by its normal vector and its local X-axis vector which are perpendicular to each other in the 3D space as shown in Fig. 1. It can be seen from Fig. 1b that the normal vector of a projection image can be determined by the Euler angles α and β , and vice versa. So, the normal vector is equivalent to the Euler angles α and β . When the normal vector is given, the local X-axis vector can be used to determine the in-plane rotation, which gives the Euler angle γ .

In the Introduction Section (Page 4, Paragraph 3), we have added the following part to show the difference between the dihedral angle and the orientations of the projections:

To represent the relative orientation between two projection images, the dihedral angle and the angle between the local X-axis vectors of the two images can be used. The dihedral angle is the angle between the normal vectors of the two images, which

is also equivalent to the distance between the two intersection points of the two normal vectors on the unit spherical surface. The relationship between the dihedral angle and projection angles (Euler angles) is shown Fig. 1b, and the formula for computing the dihedral angle from the Euler angles is also included in the Appendix. As can be seen from the Appendix, the dihedral angle is only related to the Euler angles α and β .

And in the Appendix (Page 29, Page 30), we have added the following part to show the relationship between the dihedral angles and Euler angles:

Traditionally, the 3D orientation of a projection image can be represented by Euler angles or a rotation matrix. In the case of the default right-handed system, two perpendicular vectors (the normal vector and the local X-axis vector) can be used for representing the 3D orientation as shown in Fig. 1.

Given the Euler angles α , β and γ , the orientation of a projection image can be represented by its corresponding rotation matrix:

$$\mathfrak{R}(\alpha, \beta, \gamma) = \begin{bmatrix} \cos \alpha \cos \gamma - \cos \beta \sin \alpha \sin \gamma & -\cos \beta \cos \gamma \sin \alpha - \cos \alpha \sin \gamma & \sin \alpha \sin \beta \\ \cos \gamma \sin \alpha + \cos \alpha \cos \beta \sin \gamma & \cos \alpha \cos \beta \cos \gamma - \sin \alpha \sin \gamma & -\cos \alpha \sin \beta \\ \sin \beta \sin \gamma & \cos \gamma \sin \beta & \cos \beta \end{bmatrix}$$

The normal vector of the projection image corresponds to the third column of the rotation matrix. The dihedral angle between two projection images is the angle between the normal vectors of the two projection images. Suppose that V_i and V_j are the normal vectors of the two images:

$$\begin{aligned} V_i &= (\sin \alpha_i \sin \beta_i, -\cos \alpha_i \sin \beta_i, \cos \beta_i) \\ V_j &= (\sin \alpha_j \sin \beta_j, -\cos \alpha_j \sin \beta_j, \cos \beta_j) \end{aligned}$$

The dihedral angle between the two images can be computed by:

$$\theta_{ij}^E = \arccos\left(\frac{V_i \cdot V_j}{\|V_i\| \times \|V_j\|}\right) = \arccos(\cos \beta_i \cos \beta_j + \sin \beta_i \sin \beta_j \cos(\alpha_i - \alpha_j))$$

It can be seen that the dihedral angle is only related to Euler angles α and β of the two images. Therefore, to determine all the Euler angles, two spherical embeddings are needed: the first spherical embedding determines the normal vector of each projection image, from which the Euler angles α and β can be computed; the second spherical embedding is used to compute the direction of the local X-axis vector, from which the Euler angle γ can be determined.

7. The Reviewer commented:

The authors did not show their reference maps and reconstructions determined before and after refinement of angles. It was strange appearance of the FS curves... The graphs are not consistent between different packages. So far Eman and Relion were good and providing very reasonable results. The authors did not show results of the model reconstructions. In this paper, the authors wrote that spherical embedding was not doing calculations of 3D map, that it was done in different software. How the reference map was obtained? Apparently, the authors have used a very strange band pass filtering of the original map and have distorted quite important features of the structure. The overall resolution of the map was about 25 Angstrom, the other reconstructions indicate definitely some problems with determination of angles used in the reconstructions.

[Answer]: Thank you for raising this point. For the simulate dataset, the poor result of EMAN 2.1 may be caused by that the filtering algorithm of EMAN 2.1 which is not suitable for the simulated noise. And for RELION-2, the low resolution is related to its initial model reconstruction process.

For the real dataset EMPIAR-10028, there are two important reasons for the poor results of Relion and Eman. First, in order to speed up the calculation, both the reference map EMD-2660 and the original particle images of EMPIAR-10028 have been down sampled to half of its original size. Theoretically, the resolution is only half of the origin data. Second, only the 2D class averages

are used in the reconstruction, without using any information from the original particle images. In the Section 3 (Page 19, Paragraph 1), we have added the following part:

The corresponding EM density map EMD-2660 is also downloaded, which is used as the reference map for resolution estimation. The original sampling rate for the deposited dataset of EMPIAR-10028 and EMD-2660 is 1.34 Å/pixel, which is binned by 2 in the down sampling process for our subsequent procedure, resulting in a pixel size of 2.68 Å.

In the Section 3 (Page 21, Paragraph 3), we have added the following part to explain the poor result of EMAN 2.1 and RELION-2 on the simulate dataset:

The reason why EMAN 2.1 and RELION-2 didn't perform well may be that only the 531 class averages are used in the reconstruction, without using any information from the original 11,983 images. The strategy is just set for a fair comparison, and not for exploring the full extent of the methods' capability.

In our experiments, no special filtering was used for the reference map. The low resolution of the reference map appears to be caused by down sampling.

For the reconstructions from the simulated projection images, ImageSet_A and ImageSet_B, the real map EMD-3508 serves as the reference map. For the experiments using simulate dataset, the reference map and the reconstructed initial model from 1000 projection images produced with SNR=0.2 are shown in Fig. 8 (Page 20).

Fig. 8. The reconstructions for the simulated data ($n=1000$, $SNR=0.2$). a. the reference density map (EMD-3508). Others are the reconstructed maps from the class averages by: b. the proposed method (Spherical Embedding), c. Synchronization, d. LUD, e. EMAN 2.1, and f. RELION-2.

8. The Reviewer commented:

Apparently, the authors did not realize, that nowadays in EM structural studies scientists do not use anymore classes of 2D particle images, only raw single images of particles without any averaging. That makes determination of orientations on single images significantly more challenging.

[Answer]: Thanks for your suggestion. The SNR of the real data is too low, and it is found that it is hard to reconstruct good results without 2D class averaging by the proposed method. In the future, we will try new methods to solve the problem of low SNR.

9. **The Reviewer commented:**

Please replace SPR with SPA (Single Particle Analysis (not reconstruction), which is now a broadly excepted term.

[Answer]: Thank you for raising the point. Based on your suggestion, “SPR (Single Particle Reconstruction)” has been replaced by “SPA (Single Particle Analysis)” in the revise version.

10. **The Reviewer commented:**

Typically, in EM the word “symmetricity” is not used. The standard term is “symmetry”. A protein complex may have some symmetry that typically is described by one of the point groups of symmetry.

[Answer]: Thank you for raising the point. Based on your suggestion, “symmetricity” has been replaced by “symmetry” in the revised manuscript.

11. **The Reviewer commented:**

The concept of the “embedding space” was taken from one of the cited papers, but was not explained. It is not a “natural” space. Please explain to a reader in simple terms: what is a difference between a Spherical Embedding and the Euler sphere. How can be “embedding”, which is a concept at least in 3D space, can be done on 2D surface (sphere)? This illogical link has to be explained.

[Answer]: We apologize for the unclear description. Because the orientation of a projection image can be represented as two intersection points (of the normal vector and the local X-axis vector) on

the 3D unit spherical surface. The 2D spherical surface is actually in a 3D space.

The difference between Spherical Embedding and the Euler sphere is that Spherical Embedding is a method and Euler sphere is a representation space of the orientation. In the Introduction Section (Page 4, Paragraph 2), we have added the following to explain the relationship between the results of the Spherical Embedding (the intersection points of the normal vector and local X-axis vector on the unit spherical surface) and the Euler angles:

In our method, the orientation of a projection image is represented by its normal vector and its local X-axis vector which are perpendicular to each other in the 3D space as shown in Fig. 1. It can be seen from Fig. 1b that the normal vector of a projection image can be determined by the Euler angles α and β , and vice versa. So, the normal vector is equivalent to the Euler angles α and β . When the normal vector is given, the local X-axis vector can be used to determine the in-plane rotation, which gives the Euler angle γ .

As shown in Fig. 1b, the normal vector and the local X-axis vector can be represented by their intersection points on a 3D unit spherical surface respectively. This way, the orientation of a projection image can be represented by two intersection points on the unit 3D spherical surface. It can be seen that the 3D spherical surface is a natural space for representing the orientation of the projection images.

12. The Reviewer commented:

Please explain to a reader in simple terms: why one need to have two Spherical Embedding instead of one Euler sphere? In your figure 8b no coordinates are indicated. What are they?

[Answer]: Thank you for raising the point. In our method, the orientation of a projection image can

be represented by two vectors (the normal vector and the local X-axis vector). The first spherical embedding only determines the normal vector of each projection image, in which the overall consistent dihedral angle (the angle between normal vectors) and thus the normal vectors can be derived, and then the Euler angles α and β can be computed. The second spherical embedding is used to compute the direction of the local X-axis vector from which the Euler angle γ can be determined.

Since the structure of the article has been adjusted, Fig. 8b has been renamed to Fig. 3b. Fig. 3b shows the 1000 intersection points of the normal vectors on the unit spherical surface. So, all the points in Fig. 8b are on a 3D unit spherical surface, which means that the distances from all these points to the origin are all equal to 1, and the coordinates are not necessary in this case.

Fig. 3. a. The angle between the normal vectors of two images equals to their dihedral angle in the 3D reconstruction space. N_i and N_j are the intersection points of the normal vectors of image i and image j respectively on the unit spherical surface. The distance between N_i and N_j on the unit spherical surface equals to the dihedral angle θ_{ij} . **b.** The spherical embedding of 1000 intersection points of the normal vectors on a unit spherical surface.

13. The Reviewer commented:

Please specify designation of all angles used figures 1 and 9, they were shown different, but in reality they are the same making the reader completely confused.

[Answer]: Thank you for raising the point. Since the structure of the article has been adjusted, Fig. 9 has been renamed as Fig. 4. Fig. 1 only shows how to represent the orientation of one projection image, while Fig. 4 is used to show the relative orientation between two projection images. In Fig. 1, the relationship between our representation (using the normal vector and the local X-axis vector) and Euler angles is shown. In Fig. 4, how to compute the angle between the local X-axes between two images is shown, where θ_{ij}^E is the angle between the normal vectors of the two images and φ_{ij} is the angle between the local X-axis vectors of the two images.

Fig. 1. The orientation of a projection image can be represented by its normal vector and its local X-axis vector which are perpendicular to each other. **a.** The projection image in its local coordinate system; **b.** The projection image in the 3D reconstruction space, where the normal vector can be represented by its intersection point N on the unit spherical surface, and the local X-axis vector can be represented by its intersection point A on the unit spherical surface. The Euler angle β is the angle between the Z-axis and the normal vector, the Euler angle α is the angle between the X-axis and N' vector which is the projection of the normal vector on the X-Y plane, and the Euler

angle γ is determined by the in-plane rotation of the local X-axis vector along the normal vector.

Fig. 4. Computation of the angle between the local X-axis vectors of image i and image j . Image i is on the XY plane. The common line between image i and image j is along X-axis.

14. The Reviewer commented:

Please do numbering of your formulas.

[Answer]: Thanks for your suggestion. Based on your suggestion, the numbers of formulas have been added in the revised version.

15. The Reviewer commented:

Please explain in which system of coordinates you are looking for projection orientation? It would be a tedious job and extremely time consuming to analyse all possible combinations of projections pairs if the system coordinates is not fixed (not bound to a certain orientation of the object in space). How the task of

alignment of these multiple dihedral angles (one projection N with the other $1000-1$ projections and the next $N+1$ with the other $1000-1$ projections will be aligned in space and what you will do with 1000 such spherical distributions in space? They must be adjusted somehow? Shouldn't they? What sort of information can you get out of such alignments? How is it related to the Euler orientation of each projection?

[Answer]: Thank you for raising the point. The projection orientation is determined under the unified Cartesian coordinate system which is the same Cartesian coordinate system for determining the Euler angles. In order to satisfy the independently estimated dihedral angles simultaneously, the degree of freedom has to be conserved. If there are N (>3) projection images, the independently estimated dihedral angles (angle between normal vectors) should have the same degree of freedom as the N normal vectors of the N images. Since the degree of freedom of the N normal vectors equals N , the independently estimated dihedral angles can be satisfied simultaneously in a N dimensional space, but not in a 3D space. So, by embedding the normal vectors from the N dimensional space to a 3D space, consistent dihedral angles in the three-dimensional space can be obtained.

In the Introduction Section (Page 5, Paragraph 1), we have added the following part:

However, the estimated dihedral angles are still not consistent between all the image pairs in the 3D space, because they are estimated independently. Theoretically, in order to satisfy the independently estimated dihedral angles simultaneously, the degree of freedom has to be conserved. If there are N (>3) projection images, the independently estimated dihedral angles (angle between normal vectors) should have the same degree of freedom as the N normal vectors of the N images. Since the degree of freedom of the N normal vectors equals N , the independently estimated dihedral angles can be satisfied simultaneously in a N dimensional space, but not in a 3D space. So, by reducing the dimensionality from N to 3 using a dimensionality

reduction method, the consistent dihedral angles in the 3D space can be obtained. The dimensionality reduction used in our method is spherical embedding²⁷ which is an unsupervised machine learning method whose aim is to find a representation of the data on a certain spherical surface such that the distances (or dissimilarities) between the data points are preserved as better as possible. Using the spherical embedding, the intersection points of the normal vectors of the projection images on the 3D unit spherical surface can be determined, which gives the consistent dihedral angles in the 3D space.

16. The Reviewer commented:

Please explain how the orientation of a projection image (three coordinates of one projection: alpha, beta, gamma) can be represented by two points on the spherical surface, and the global consistency constraints? Where the two points are coming from for one projection? Where are they located and what is a mathematical link between them?

[Answer]: Thank you for raising the point. In our method, the orientation of a projection image is represented by its normal vector and its local X-axis vector which are perpendicular to each other. As shown in Fig. 1b, the normal vector and the local X-axis vector can be represented by their intersection points on a 3D unit spherical surface respectively. Since the degree of freedom for each intersection point on the 3D unit spherical surface is 2, and the distance between the two intersection points should be $\pi/2$ (representing the two perpendicular vectors) which sacrifices 1 degree of freedom, the degree of freedom for the two intersection points is 3. Therefore, the representation using the two intersection points is equivalent to the Euler angle representation which has 3 independent variables (the degree of freedom is also 3).

The mathematical link between Euler angles and the normal vector is the same as the coordinate transformation formula between the spherical coordinate and the Cartesian coordinate:

$$\begin{bmatrix} x_i \\ y_i \\ z_i \end{bmatrix} = \begin{bmatrix} \sin \beta_i \cos \alpha_i \\ \sin \beta_i \sin \alpha_i \\ \cos \beta_i \end{bmatrix}$$

where (x_i, y_i, z_i) are the coordinates of the normal vector in the Cartesian coordinate system, and α_i and β_i are the Euler angles.

The mathematical link between Euler angles and the local X-axis vector is shown by the Formula 12 (Page 13):

$$\begin{bmatrix} \cos \gamma_i \\ \sin \gamma_i \\ \delta_i \end{bmatrix} = \begin{bmatrix} \cos \beta_i & 0 & -\sin \beta_i \\ 0 & 1 & 0 \\ \sin \beta_i & 0 & \cos \beta_i \end{bmatrix} \cdot \begin{bmatrix} \cos \alpha_i & \sin \alpha_i & 0 \\ -\sin \alpha_i & \cos \alpha_i & 0 \\ 0 & 0 & 1 \end{bmatrix} \cdot \begin{bmatrix} \sin \beta_i'' \cos \alpha_i'' \\ \sin \beta_i'' \sin \alpha_i'' \\ \cos \beta_i'' \end{bmatrix}$$

17. The Reviewer commented:

Part with methods should go before the section with results.

[Answer]: Thanks for your suggestion. Based on your suggestion, the method part has been moved to Section 2 before the result part.

Response to Reviewer 2

We very much appreciate the comments and suggestions from Reviewer 2, which help to improve the quality and description of the manuscript. We have revised the manuscript according to the comments of Reviewer 2, where all changes have been highlighted in the manuscript. In the following text, the point-by-point replies are summarized as follows:

1. The Reviewer commented:

This is a fine and interesting manuscript. It leverages the understanding of the geometry of projections to utilize algorithms involving data on the sphere (Spherical Embedding). The performance of the algorithm seems to be a significant improvement in assigning angles to projections over other programs (EMAN, relion).

[Answer]: Thanks for the positive comments!

2. The Reviewer commented:

I have some minor grammatical criticisms. I would replace symmetricity with symmetry, where it appears.

[Answer]: Thank you for raising the point. “symmetricity” has been replaced by “symmetry” in the revised version.

3. **The Reviewer commented:**

Section 2.1.1

"which are called as the " -> "which are called the".

[Answer]: Thanks for your suggestions. Because the structure of the article has been adjusted, the sentence “which are called as the reference angles” has been modified to “which are called the reference angles” in Subsection 3.1.1 (Page 14, Paragraph 1).

Response to Reviewer 3

We very much appreciate the comments and suggestions from Reviewer 3, which help to significantly improve the quality and description of the manuscript. We have revised the manuscript according to the comments of Reviewer 3, where all changes have been highlighted in the manuscript. In the following text, the point-by-point replies are summarized:

1. The Reviewer commented:

The manuscript of Lu and colleagues offers a refined approach on an important theoretical aspect in cryo-electron microscopy (cryo-EM): calculating the initial orientation of 2D images for reconstructing a 3D object.

.....

The article is very well written, the results are clearly organized. The authors provide an exemplar description of the methodology: all the definitions are provided, the rationale of each step is clearly stated, and the experiments are clearly described.

[Answer]: Thanks for the positive comments!

2. The Reviewer commented:

However, I have to say, the article is rather theoretical and the 'real images' test is limited to a ribosome data set. Without a richer palette of relevant biological examples of different sizes and complexity.

[Answer]: Thanks for your suggestions. Based on your suggestion, we have added the experimental results for a new real dataset EMPIAR-10328. The original data (particle images and star file) are downloaded from the EMPIAR database (EMPIAR-10328). After 2D class averaging, five algorithms are used to do the initial model reconstruction. The comparison of the reconstruction results is displayed in Fig. 12, and the evaluation of the reconstruction results is shown by the FSC curves in Fig. 11. The new experimental part has been added in Section 3.2.2 (Page22-Page24).

Fig. 11. a. Comparison of the FSC curves produced by the proposed method (Spherical Embedding), Synchronization, LUD, EMAN 2.1, and RELION on the EMPIAR-10328 dataset. b. The zoom-in region of the red dashed rectangle in a, which shows the details of the FSC curves around 0.12 1/Å.

Fig. 12. The reconstructions for EMPIAR-10328. a. the reference density map (EMD-22689). Others are the reconstructed maps from the class averages by: b. the proposed method (Spherical Embedding), c. Synchronization, d. LUD, e. EMAN 2.1, and f. RELION-2.

3. The Reviewer commented:

Without a richer palette of relevant biological examples of different sizes and complexity, the manuscript would be a better fit for a specific journal, such as Journal of Structural Biology.

[Answer]: Thank you for the suggestion! We will consider the Journal of Structural Biology for our future work in the 3D reconstruction.

4. **The Reviewer commented:**

There is a major issue which I would like to be addressed by the authors. Finding the correct orientation of a common line between two images is strongly dependent on the centering of each particle. At no point in the manuscript do the authors indicate any concern regarding this point. For the simulated images, the authors do just rotate the map in order to generate the artificial projections already centered. For the 'real images' case, they use 2D averages and not original images – a reasonable way to deal with the signal to noise ratio. As they do not mention particle picking, I suppose they used directly the EMPIAR data which raises the question how accurate was the centering and how dependent is their method of the variations of this parameter.

[Answer]: We apologize for the unclear description. In the proposed method, after calculating the 2D class averages of the real images, the centers of the 2D class averages are aligned using a center alignment algorithm from ASPIRE 0.14 toolkit. This center alignment algorithm is also used before using Synchronization and LUD. As for the RELION-2 and EMAN 2.1, there is a center alignment in their own initial model reconstruction algorithm. In the Section 3 (Page 20, Paragraph 1), we have added the following:

For Spherical Embedding, Synchronization and LUD, the class averages are aligned by the center alignment algorithm from ASPIRE 0.14 toolkit. While for EMAN 2.1 and RELION-2, they have their own center alignment algorithm in the initial model reconstruction process.

REVIEWERS' COMMENTS:

Reviewer #1 (Remarks to the Author):

The MS has been improved and is more coherent. However, the authors used the wrong numbers for SNR. That must be corrected ASAP. The authors have reversed the definition: if SNR is equal to 1, it means that noise is very strong (see the definition below), and the error in refining of angles should be much higher compared to the SNR = 0.1, where the signal is much stronger making the search of angles much easier having lower errors. That have been shown both in Figure 5 and Table 1 but in an absolutely wrong order. The parameters of SNR have to be in the opposite order : better results at the lower SNR = 0.1 and the worst are at the SNR = 1, where intensity of noise is the same as the signal. See below:

Definition of SNR:

Signal-to-noise ratio (SNR or S/N) is a measure used in science and engineering that compares the level of a desired signal to the level of background noise

(https://www.google.com/search?client=firefox-b-d&q=definition+of+snr*; In other words, SNR is the ratio of signal power to the noise power

$$\text{SNR} = \text{Power signal} / \text{Power Noise} = (\text{Amplitude signal(RMS)} / \text{Amplitude Noise(RMS)})^2$$

Reviewer #3 (Remarks to the Author):

The authors have considerably improved the quality of the manuscript and I recommend the publication.

Response to Referees

We would like to thank all the referees for their valuable comments which have helped us a lot in improving the manuscript. All the issues raised by the reviewers have been carefully considered and addressed. In the following, our detailed responses (in blue color) to every concern raised by the referees are listed:

Response to Reviewer 1

1. **The Reviewer commented:**

The MS has been improved and is more coherent.

[Answer]: Thanks for the positive comment!

2. **The Reviewer commented:**

However, the authors used the wrong numbers for SNR. That must be corrected ASAP. The authors have reversed the definition: if SNR is equal to 1, it means that noise is very strong (see the definition below), and the error in refining of angles should be much higher compared to the SNR = 0.1, where the signal is much stronger making the search of angles much easier having lower errors. That have been shown both in Figure 5 and Table 1 but in an absolutely wrong order. The parameters of SNR have to be in the opposite order : better results at the lower SNR = 0.1 and the worst are at the SNR = 1, where intensity of noise is the same as the signal. See below:

Definition of SNR:

Signal-to-noise ratio (SNR or S/N) is a measure used in science and engineering that compares the level of a desired signal to the level of background noise

([https://www.google.com/search?client=firefox-b-d&q=definition+of+snr](https://www.google.com/search?client=firefox-b-d&q=definition+of+snr*);*

In other words,

SNR is the ratio of signal power to the noise power

SNR = Power signal/Power Noise = (Amplitude signal(RMS)/Amplitude Noise(RMS))^2

[Answer]: There may be some misunderstandings about the SNR. As given by the reviewer, the definition of SNR is:

$$\text{SNR} = \frac{\text{Power of Signal}}{\text{Power of Noise}}$$

According to the definition, the smaller the SNR, the stronger the noise power, because the signal power is the same for the same simulated dataset. Therefore, when $\text{SNR} = 1$, the noise power of the image is lower than that when $\text{SNR} = 0.1$, and the error in refining of angles should be lower compared to $\text{SNR} = 0.1$. So, the original order of SNR in Fig. 5 and Table 1 is correct.

Response to Reviewer 3

1. **The Reviewer commented:**

The authors have considerably improved the quality of the manuscript and I recommend the publication.

[Answer]: Thanks for the positive comment!